



# Evaluation of model-derived root-zone soil moisture over the Huai river basin

En Liu[1,2], Yonghua Zhu[1], Jean-christophe Calvet[2], Haishen Lü[1], Bertrand Bonan[2], Jingyao Zheng[1], Qiqi Gou[1], Xiaoyi Wang[1], Zhenzhou Ding[1], Haiting Xu[1], Ying Pan[1], Tingxing Chen[1]

[1]State Key Laboratory of Hydrology-Water Resources and Hydraulic Engineering, 14 College of Hydrology and Water Resources, Hohai University, Nanjing 210098, China
[2]CNRM, Université de Toulouse, Météo-France, CNRS, 31057, Toulouse, France

*Correspondence to*: Yonghua Zhu (zhuyonghua@hhu.edu.cn)

**Abstract:** Root-zone soil moisture (RZSM) is crucial for water resource management, drought monitoring and sub-seasonal flood climate forecast. RZSM is not directly observable from space but various model-derived RZSM products are available at the global scale and are widely used. In this paper, a comprehensive quantitative evaluation of eight RZSM products is made over the Huai river basin (HRB) in China. A direct validation is performed using observations from 58 in situ soil moisture stations from 1 April 2015 to 31 March 2020. Attention is drawn to the potential factors increasing uncertainties of model-generated RZSM, such as errors on atmospheric forcings (precipitation, air temperature), soil properties, and model parameterizations. Results indicate that the Global Land Data Assimilation System Catchment Land Surface Model (GLDAS_CLSM) performs best among all RZSM products with the highest correlation coefficient (R) and lowest unbiased root-mean square error (ubRMSE): 0.503 and 0.031 $m^3\,m^{-3}$, respectively. All RZSM products tend to overestimate the in situ soil moisture values, except for the Soil Moisture and Ocean Salinity (SMOS) L4 product, which underestimates RZSM. The underestimated SMOS L3 SSM associated with low physical surface temperature triggers the underestimation of RZSM in SMOS L4. The RZSM overestimation by other products can be explained by the overestimation of precipitation amount, precipitation event frequency (drizzle effects) and by the underestimation of air temperature. Besides, the overestimation of the soil clay content and the underestimation of the soil sand content in different LSMs leads to larger soil moisture values. The intercomparison of the eight RZSM products shows that MERRA-2 and SMAP L4 RZSM are the most correlated with one another. These products are based on the same LSM and on the same surface meteorological forcing generated from the National Aeronautics and Space Administration (NASA) GEOS-5. In addition, model parameterizations in different LSMs vary considerably, affecting the transfer and exchange of water and heat in the vadose zone.



## 1 Introduction

Soil moisture plays a key role in the hydrological cycle and in land-atmosphere interactions. It controls the
water and energy balances (Calvet, 2000, Brocca et al., 2010, Xing et al., 2021), and has been recognized as one
of the 50 essential climate variables by the World Meteorological Organization (WMO) (Cho et al., 2015). In
particular, the root-zone soil moisture (RZSM, 0-100 cm) has important applications in agricultural drought
monitoring, water resources management, flood prediction and seasonal climate forecast (Reichle et al., 2017,
Zhou et al., 2020, Beck et al., 2021). In the context of climate change, extreme events (floods and droughts,
heatwaves, etc.) affecting RZSM tend to occur more frequently around the world (Lorenz et al., 2010, Hauser et
al., 2016, Al Bitar et al., 2021). For example flash droughts affect, more and more, the Huaibei plain in China
(Gou et al., 2022).
Recent satellite soil moisture missions provide global, ~3-day resolution soil moisture retrievals limited to
the top few centimeters (0-5 cm for L band) due to the limitation of microwave penetration depth (Bi et al., 2016).
So various model-derived RZSM products are developed from wider global scale applications. For example,
model-based products such as the Global Land Data Assimilation System (GLDAS), based on the GLDAS_NOAH
and on the GLDAS Catchment land surface models (GLDAS_CLSM) (Bi et al., 2016), the China Land Data
Assimilation System (CLDAS) (Shi et al., 2014), and Soil Moisture Active Passive (SMAP) Level 4 (L4)
(Rienecker et al., 2008, Reichle et al., 2017), were developed. They aim to provide the optimal land surface states
and fluxes through the combination of an offline (not coupled to the atmosphere) Land Surface Model (LSM) and
satellite data by data assimilation techniques (Calvet and Noilhan, 2000, Rodell et al., 2004). The LSM is forced
with meteorological analysis fields (precipitation, wind speed, air humidity, surface pressure, air temperature and
radiance). Moreover, the European Centre for Medium-Range Weather Forecasts (ECMWF) fifth generation
reanalysis (ERA5) (Albergel et al., 2018), the Modern-Era Retrospective Analysis for Research and Applications
version 2 (MERRA-2) (Gelaro et al., 2017) and the National Centers for Environmental Prediction Climate
Forecast System Version 2 (NCEP CFSv2) (Saha et al., 2014) also provide global, subdaily/daily resolution
analysis fields of atmosphere, ocean and land surface variables through coupling an atmospheric general
circulation model (AGCM) with a LSM and an Ocean Wave Model (OWM) as well as assimilating large amounts
of in situ and satellite-derived observations (Saha et al., 2014, Reichle et al., 2017). Soil Moisture and Ocean
Salinity (SMOS) Centre Aval de Traitement des Données (CATDS) provides SMOS L4 RZSM derived from
SMOS Level 3 (L3) 3-day SSM using a statistical exponential filter model (Albergel et al., 2008).



Large amounts of studies were conducted to validate and assess the utility of SSM using in situ observations
in the topsoil layer (Collow et al., 2012, Cui et al., 2017, Beck et al., 2021, Zheng et al., 2022), more rarely for
RZSM, especially in China (Xing et al., 2021, Xu et al., 2021). Being one of the important agricultural grain
production areas in China, it is crucial to assess the performance of various RZSM products over the Huai River
Basin (HRB). Model-derived RZSM products are commonly validated using in situ observations, which can be
considered as the reference data set with highest quality. Differences between in situ and model-derived RZSM
may be caused by errors in the model meteorological forcing data, soil properties, parameterization, and by the
scale mismatch. Nevertheless, using in situ observations may be the most accurate method for soil moisture
validation (Xu et al., 2021). Many studies have evaluated the satellite-derived SSM or model-derived RZSM using
in situ soil moisture observations (Albergel et al., 2012, Cui et al., 2017, Reichle et al., 2017, Pablos et al., 2018,
Beck et al., 2021, Wang et al., 2021, Xing et al., 2021, Xu et al., 2021). Further, Rüdiger et al. (2009) made the
intercomparison of different SSM products with one other together with the comparison with in situ soil moisture
observations.
The quality of meteorological forcing data (mainly precipitation and air temperature) is one of the most
important factors determining the accuracy of model-derived RZSM simulations (Zeng et al., 2021). However,
numerous studies showed that there exist large uncertainties in atmospheric forcing data derived from global
climate model, in particular, the precipitation frequency, intensity and heavy precipitation events (Sun et al., 2005,
Piani et al., 2010, Velasquez et al., 2020, Jiao et al., 2021). Describing soil properties right is also important. Many
global LSMs use the FAO/UNESCO (Food and Agriculture Organization, United Nations Educational, Scientific
and Cultural Organization) soil map of the World generated in 1981, for instance, GLDAS products (Bi et al.,
2016, Yang et al., 2020), NCEP CFSv2 (Yang et al., 2020), ERA5 (Qin et al., 2017, Yang et al., 2020), SMOS L4
(Al Bitar et al., 2021), MERRA-2 (Koster et al., 2016, Gelaro et al., 2017) and SMAP L4 (Reichle et al., 2019),
which incorporates little soil information in many regions including China (Shangguan et al., 2013). This increases
the uncertainty of soil moisture simulations. Moreover, soil stratification may influence RZSM. In the Huaibei
plain, the plough, black soil and lime concretion layers stratification may impede the vertical transfer of water
from the surface layer to the root-zone layer. Finally, the quality of the model parameterizations are key factors
determining the accuracy of soil moisture simulations. Different LSMs are used in LDAS or reanalysis products,
such as the Noah LSM in GLDAS_NOAH and NCEP CFSv2 (Rodell et al., 2004, Saha et al., 2014), HTESSEL
in ERA5 (Yang et al., 2020), CLSM in GLDAS_CLSM, MERRA-2 and SMAP L4 (Koster et al., 2000, Reichle
et al., 2017, Reichle et al., 2019), the Community Land Model 3.5 (CLM), Common Land Model (CoLM) and the


community Noah land surface model with multi-parameterization options (Noah-MP) in CLDAS products (Wang
et al., 2021). The exponential filter technique is used in SMOS L4 (Al Bitar et al., 2021).

The objectives of this study are as follows: (1) compare eight global RZSM products (ERA5, MERRA-2,

NCEP CFSv2, GLDAS_CLSM v2.2, GLDAS_NOAH v2.1, CLDAS v2.0, SMAP L4 and SMOS L4) with in situ
soil moisture observations over HRB from 1 April 2015 to 31 March 2020, (2) intercompare the RZSM products
with one another over HRB, (3) investigate the potential error sources of RZSM (meteorological forcing data, soil
properties and soil stratification, model parameterizations).



## 2 Datasets

### 2.1 HRB in situ measurements

The HRB is the transitional zone between northern subtropical and warm temperate climates and one of the most important commodity grain production areas in China. It is located in eastern China, 111°55′-121°25′ E, 30°55′-36°36′ N, and covers an area of 270000 km$^2$ (Figure 1). The HRB has a typical humid and sub-humid monsoon climate. The average annual precipitation is 888 mm and increases from north to south. More than 60% of the annual precipitation occurs in four months, from June to September (Zhang et al., 2009). The annual evaporation ranges from 900 to 1500 mm and decreases from north to south. The HRB suffers from frequent floods and droughts due to the spatiotemporal variability of precipitation and evaporation. The main land cover types over HRB are rainfed croplands, followed by irrigated croplands, then woodlands and grasslands. Overall, the terrain of HRB is relatively flat, a large plain accounting for 90% of the area of the whole HRB.

The HRB soil moisture network was deployed by the Ministry of Water Resources of the People's Republic of China. It consists of 58 in situ stations and provides soil moisture measurements at 4 depths of 10 cm, 20 cm, 40 cm and 100 cm. At each station, volumetric soil moisture measurements in unit of $m^3\,m^{-3}$ are collected at 08:00 AM local solar time using Frequency Domain Reflectometry ECH$_2$O EC-TM probes. These probes are calibrated using gravimetric measurements sampled at four soil depths. The soil moisture measurements are quality controlled for filtering out unreliable data before using them for validating model-derived RZSM products. Among the 58 stations, 51 stations are located in the relatively flat Huaibei plain, mainly covered by rainfed crops, 5 stations are located in the irrigated cropland area and 2 stations are located in the woodland area. Since this study aims to evaluate the accuracy of model-derived RZSM products (0-100 cm), the soil moisture measurements at 4 depths are depths-weighted averaged for obtaining the 0-100 cm soil moisture data.

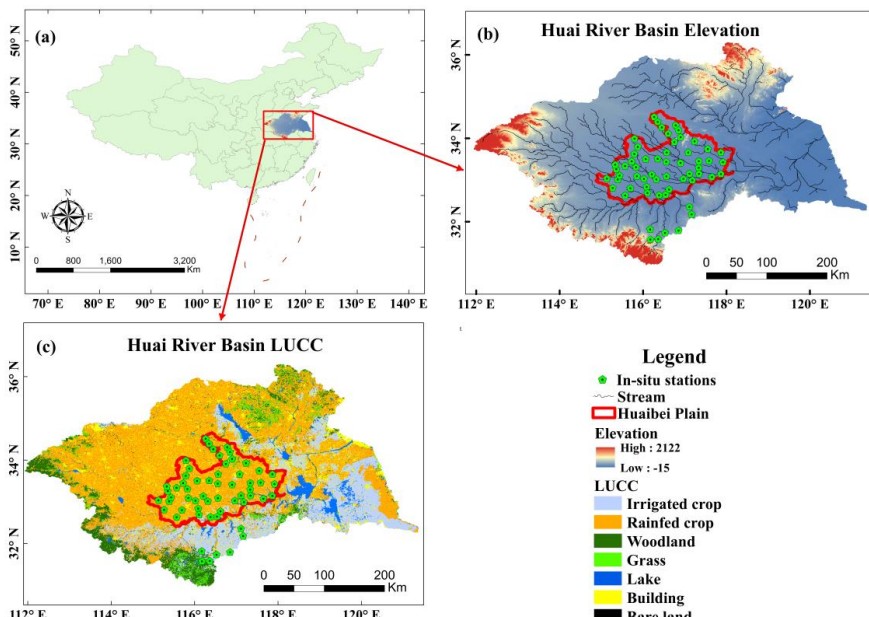

**Fig. 1 Location of the study area and distribution of in-situ soil moisture stations. Fig. 1 (c) shows the land cover types of Huai River Basin (HRB) where the in situ stations are mainly covered by rainfed crop**.

China daily ground rainfall and air temperature gridded dataset V2.0 is provided by China Meteorological Administration (CMA) (http://data.cma.cn) at a spatial resolution of 0.5°×0.5°. These data are used to validate the meteorological forcing fields used in reanalysis and LDAS. The CMA gridded dataset is obtained by interpolating spatially using the method of partial thin-plate smoothing splines from 2474 national ground meteorological station observations after quality controls and corrections. The average coverage rate of gauging stations located in a grid cell is 38% across the whole China, but up to 77% in eastern part of China where the HRB is located. The dataset was comprehensively validated and has high quality. The rainfall data has mean RMSE of 0.49 mm/month and R of 0.93 significant at $p < 0.01$ (CMA, 2012). The mean yearly air temperature data has a mean bias of $\pm 0.2°C$ and RMSE of 0.2-0.3°C (CMA, 2012).

**2.2 Soil map**

Currently, soil databases used in many global LSMs are derived from the FAO/UNESCO soil map of the World at 1:5 million scale. It took twenty years to complete this map which remained until recently the only global overview of soil resources (Shangguan et al., 2013). However, this soil map incorporated little soil information in many regions including China. Given these uncertainties of in soil properties, the variables simulated by LSMs



(e.g., RZSM) presented larger errors over China (Nachtergaele et al., 2009, Shangguan et al., 2013). Hence, the
Harmonized World Soil Database (HWSD) with a resolution of 30 arc-second was produced by FAO and the
International Institute for Applied Systems Analysis (IIASA) by combing recently collected regional and national
updates of soil information with the FAO/UNESCO soil map of the world at 1:5 million scale. HWSD includes
the soil map of China provided by the Institute of Soil Science, Chinese Academy of Sciences (ISSCAS) at 1:1
million scale.
The soil data set developed by Shangguan et al. (2013) is used in the CLDAS (Qin et al., 2017), which
integrates the physical and chemical attributes of 8979 soil profiles and the Soil Map of China (Shangguan et al.,
2013). The data set contains soil properties information for eight layers (0-2.3 m) at the spatial resolution of 30×30
arc-seconds. Due to the lack of the measured soil data, the soil properties information (sand and clay content, bulk
density and soil organic matter) obtained from Shangguan et al. (2013) was used to validate the accuracy of that
from FAO/UNESCO and HWSD.
**2.3. Model-derived RZSM products**
**2.3.1 ERA5**
ERA5 is the ECMWF fifth generation atmospheric reanalysis of the global climate and weather. It covers the
period from January 1950 to present, and substitutes for the ERA-Interim reanalysis. ERA5 is developed using 4-
Dimensional Variational (4D-Var) data assimilation with an underlying 10-member ensemble and model forecasts
in CY41R2 of the ECMWF Integrated Forecast System (IFS), with 137 hybrid sigma/pressure model levels in the
vertical and the top level at 0.01 hPa (Xu et al., 2021). The temporal and spatial resolutions of ERA5 dataset are 1
hour and 31 km (regridded to a regular lat-lon grid of 0.25 degree), respectively. The 4D-Var data assimilation
uses 12 hour windows from 0900 UTC to 2100 UTC and from 2100 UTC to 0900 UTC (the following day)
(Albergel et al., 2018).
**2.3.2 MERRA-2**
MERRA-2 is the latest version of global atmospheric reanalysis for the satellite era produced by NASA Global
Modeling and Assimilation Office (GMAO) using an upgraded version of Goddard Earth Observing System Model
(GEOS-5) and the Gridpoint Statistical Interpolation assimilation system (Reichle et al., 2017). Owing to the fact
that the MERRA data assimilation system was set in 2008 and could not integrate new data types, MERRA-2 was
developed. In comparison with the MERRA reanalysis, MERRA-2 contains many updates and new fundamental
developments in modeling and 3D-VAR data assimilation. It assimilates aerosol observations and other new



observational forcings enabling the land surface model to provide more stable land feedback processes (Gelaro et
al., 2017). Moreover, the Climate Prediction Center (CPC) Unified Gauge-Based Analysis of Global Daily
Precipitation (CPCU) product and the CPC Merged Analysis of Precipitation (CMAP) product from the National
Oceanic and Atmospheric Administration (NOAA) CPC are used in MERRA-2 precipitation corrections, which
allows the observed precipitation to impact, via evapotranspiration, the near-surface air temperature and humidity,
thereby yielding a more self-consistent near-surface meteorological dataset (Reichle et al., 2017). The dataset
covers the period from 1980 to present with a latency of ~3 weeks after the end of a month and has a temporal
resolution of 1 hour and spatial resolution of $0.5° \times 0.625°$. The dataset was regridded to GLDAS-2_0.25 through
bilinear interpolation with a regular latitude-longitude grid of 0.25 degree.

### 2.3.3 NCEP CFSv2

NCEP CFSv2 is a global, high resolution, coupled atmosphere-ocean-land surface-sea ice system designed
to provide the best estimate of the state of these coupled domains. The Noah land surface model is used in both
the coupled land surface-atmosphere-ocean model, and in the Global Land Data Assimilation System (GLDAS)
(Saha et al., 2014). Compared to NCEP reanalyses 1 and 2 (R1, R2), CFSv2 involves several upgrades: improved
forecast model and data assimilation scheme, finer spatial resolution, assimilation of satellite radiances rather than
retrievals, simulation of four soil levels (0-10 cm, 10-40 cm, 40-100 cm and 100-200 cm) rather than two soil
levels (0-10 cm and 10-200 cm) (Lu et al., 2005).

### 2.3.4 GLDAS_NOAH

GLDAS_NOAH Version 2.1 provides global, 3-hourly, 0.25-degree resolution of estimates covering the
period from 1 January 2000 to present. The Noah land surface model simulates four soil levels, including 0-10 cm,
10-40 cm, 40-100 cm, 100-200 cm and uses the Modified IGBP MODIS 20-category vegetation classification and
the soil properties based on the Hybrid STATSGO/FAO datasets (Bi et al., 2016). GLDAS drives the Noah model
by ingesting observation-based data NOAA/Global Data Assimilation System (GDAS) atmospheric analysis fields,
the disaggregated Global Precipitation Climatology Project (GPCP) V1.3 Daily Analysis precipitation fields and
the Air Force Weather Agency's AGRicultural METeorological modeling system (AGRMET) radiation fields)
(Rui et al., 2021).

### 2.3.5 GLDAS_CLSM

GLDAS_CLSM Version 2.2 is based on the CLSM forced with the meteorological analysis fields from the
operational ECMWF Integrated Forecasting System (Rui et al., 2021). The Catchment model uses the Mosaic land





cover classification, together with soils, topographic, and other model-specific parameters that are derived in a
manner consistent with that of the GEOS-5 climate modeling system. Alternatively, the Daily Catchment model
simulations use the University of Maryland (UMD) land cover classification, with the rest of parameters from the
GEOS-5 system. Compared with GLDAS-2.0 and GLDAS-2.1 (open-loop, i.e., no data assimilation), GLDAS-
2.2 assimilates the total terrestrial water anomaly observations from Gravity Recovery and Climate Experiment
(GRACE). GLDAS_CLSM 2.2 provides global, daily, 0.25-degree resolution estimates covering the period from
1 February 2003 to present.
**2.3.6 CLDAS**
The CLDAS-2.0 product is developed and released by CMA based on a multi-LSMs operational system
consisting of CLM, CoLM, and Noah-MP, with a spatial coverage of 0-60° N and 70-150° E. The production of
CLDAS-V2.0 includes the following three processes. Firstly, nearly 40000 automatic meteorological stations
measurements, ECMWF and NCEP numerical analysis/forecast product, satellite-derived precipitation (FY2) and
Digital Elevation Model (DEM) are used to produce 0.0625°, hourly estimates of meteorological forcing data by
operating the Space-Time Multi-Scale Analysis System (STMAS) (Shi et al., 2014, Wang et al., 2021). Meantime,
the meteorological forcing is validated using national automatic station observations (more than 2400 stations).
Secondly, the meteorological forcing is used to drive the multi-LSMs system for obtaining a multilayer soil
moisture estimates ensemble. Finally, ensemble-average is applied to each soil layer to generate a soil moisture
ensemble analysis product.
**2.3.7 SMAP L4**
The SMAP Level-4 soil moisture (L4-SM) is produced by assimilating SMAP radiometer level-1C brightness
temperature observations into CLSM and provides global, 3-hourly, 9-km resolution estimates of SSM (0-5 cm)
and RZSM (0-100 cm) (Reichle et al., 2019). The Goddard Earth Observation System, version 5, LDAS (GEOS-
5 LDAS) is based on a spatially distributed ensemble Kalman filter (EnKF) and CLSM (Rienecker et al., 2008).
The GEOS-5 CLSM is driven by surface meteorological data (precipitation, radiation, etc.) from GEOS-5 Forward
Processing (FP) system. Large amounts of observations are assimilated into a global atmospheric model and CPCU,
0.5-degree, daily precipitation observations are used for correcting the GEOS-5 precipitation. The EnKF has a 3-
hourly update time step and is used to interpolate and extrapolate the brightness temperature and model estimates
in time and space (Reichle et al., 2017).





### 2.3.8 SMOS L4

The SMOS L4 soil moisture product is produced by SMOS CATDS and provides global, daily estimates of RZSM (0–100 cm) over a 25-km EASE-2 grid from January 2010 to present. The SMOS L4 RZSM is derived from SMOS L3 3-day SM product (descending orbit, 06:00 PM) and other ancillary datasets, such as MODIS observations and climate data from the NCEP and an upgraded FAO/UNESCO soil properties map, using a modified exponential filter linking the characteristic time length T (the transfer time for water from surface layer to root zone layer) to the soil properties (Pablos et al., 2018). The soil column is divided into three layers (layer1: 0-5 cm, layer2: 5-40 cm, layer3: 40-100 cm) in a water bucket model. The scaled 0-5 cm soil moisture is modified using a logarithmic function and applied to the water bucket model to obtain 5-40 cm soil moisture combined with T1 from layer1 to layer2. Then the scaled 5-40 cm soil moisture and T2 from layer2 to layer3 are applied to the water bucket model to obtain 40-100 cm soil moisture. Finally, the RZSM (0-100 cm) is computed based on a depth-weighted average of the three layers' soil moisture (Al Bitar et al., 2021).

The eight model-derived RZSM products evaluated in this study are summarized in Table 1.





Table 1 Description of global (regional) RZSM products from model-based land surface states in the study.

| Dataset | Land surface model | Time period | Temporal resolution | Spatial resolution | Soil layers | Data access |
|---|---|---|---|---|---|---|
| ERA5 (Global) | HTESSEL | January 1, 1979-present | Hourly | 31km×31km (0.25°×0.25° regridded) | 0-7 cm, 7-28 cm, 28-100 cm, 100-289 cm | ERA5 reanalysis datasets Hourly 0.25 x 0.25 degree\| ECMWF |
| MERRA-2 (Global) | CLSM | January 1, 1980-present | Hourly | 0.5°×0.625° (0.25°×0.25° regridded) | 0-5 cm, 0-100 cm | GES DISC Dataset: MERRA-2 tavg1_2d_lnd_Nx (M2T1NXLND 5.12.4) (nasa.gov) |
| NCEP CFSv2 (Global) | Noah | January, 2011-present | 6-Hourly | 0.20°×0.20° | 0-10 cm, 10-40 cm, 40-100 cm, 100-200 cm | CISL RDA: NCEP Climate Forecast System Version 2 (CFSv2) 6-hourly Products (ucar.edu) |
| GLDAS_NOAH (Global) | Noah | January 1, 2000-present | 3-Hourly | 0.25°×0.25° | 0-10 cm, 10-40 cm, 40-100 cm, 100-200 cm | GES DISC Dataset: GLDAS Noah Land Surface Model L4 3 hourly 0.25 x 0.25 degree V2.1 (nasa.gov) |
| GLDAS_CLSM (Global) | CLSM | February 1, 2003-present | Daily | 0.25°×0.25° | 0-2 cm, 0-100 cm | GES DISC Dataset: GLDAS Catchment Land Surface Model L4 daily 0.25 x 0.25 degree GRACE-DA1 V2.2 (nasa.gov) |
| CLDAS (Asia) | CLM CoLM Noah-MP | January 1, 2008-present | Hourly | 0.0625°×0.0625° | 0-5 cm, 0-10 cm, 10-40 cm, 40-100 cm, 100-200 cm | China Meteorological Administration Land Data Assimilation System (CLDAS v2.0) Product Dataset (cma.cn) |
| SMAP Level 4 (Global) | CLSM | March 31, 2015-present | 3-Hourly | 9 km×9 km | 0-5 cm, 0-100 cm | SMAP L4 Global 3-hourly 9 km EASE-Grid Surface and Root Zone Soil Moisture Analysis Update, Version 5 \| National Snow and Ice Data Center (nsidc.org) |
| SMOS Level 4 (Global) | Exponential filter (no LSM) | January 14, 2010-present | Daily | 0.25°×0.25° | 0-100 cm | L4 Land research products - Centre Aval de Traitement des Données SMOS (CATDS) |




**3 Methods**
**3.1 Statistical metrics**
Four widely used statistical metrics were used to quantitatively evaluate the performance of RZSM products
against in situ measurements. The Pearson correlation coefficient (R) measures the degree of linear correlation
between the in situ measurements and model-derived RZSM, Mean Bias Error (MBE) reflects the mean systematic
deviation of model simulations relative to the measurements, Root Mean Square Error (RMSE) and ubRMSE
measure standard deviation of random error (Zheng et al., 2022). In addition, Probability of Detection (POD),
False Alarm Ratio (FAR) and Critical Success Index (CSI) are used to assess the ability of model-derived rainfall
to reproduce the measured rainfall (Su et al., 2019). The statistical metrics and corresponding formulas are listed
in Table 2.
**3.2 Calculation and validation of RZSM**
Since the in situ measurements are available at several specific depths (10 cm, 20 cm, 40 cm and 100 cm),
the RZSM is calculated with a depth-weighted average of the four layers soil moisture. The equation is as follows:
$\theta_{RZSM} = \frac{2\theta_1 L_1 + (\theta_1+\theta_2)L_2 + \cdots (\theta_{n-1}+\theta_n)L_n}{2(L_1+L_2+L_3+\cdots L_n)}$ (1)
where $\theta_{RZSM}$ refers to the RZSM in the 0-100 cm ($m^3 m^{-3}$), $\theta_n$ is the volumetric soil moisture at the $n_{th}$ observation
depth ($m^3 m^{-3}$), and $L_n$ is the soil layer thickness between adjacent observation depths (m).
For the model-derived RZSM products, apart from the GLDAS_CLSM, MERRA-2, SMAP L4 and SMOS
L4 directly providing the 0-100 cm RZSM, other RZSM products are provided in different soil layers, NCEP
CFSv2, CLDAS and GLDAS_NOAH ($\theta_{0-10\ cm}$, $\theta_{10-40\ cm}$, $\theta_{40-100\ cm}$), ERA5 ($\theta_{0-7\ cm}$, $\theta_{7-28\ cm}$, $\theta_{28-100\ cm}$).
For instance, the GLDAS_NOAH RZSM can be calculated as:
$\theta_{RZSM} = 0.1 \times \theta_{0-10\ cm} + 0.3 \times \theta_{10-40\ cm} + 0.6 \times \theta_{40-100\ cm}$ (2)
In this study, the model-derived soil moisture is directly compared with point-scale observations for each
station located within the model grid cell. If there are more than one in-situ station in a grid cell, the average soil
moisture observations of all stations in a grid cell is used to compare with model-derived grid value.





Table 2 List of the statistic metrics for evaluating RZSM products and corresponding precipitation forcing data
using in situ measurements.

| Statistic metrics | Unit | Equation | Optimal |
|---|---|---|---|
| correlation coefficient (R) | - | $R = \dfrac{\sum_{i=1}^{n}\left(\theta_{est,i} - \overline{\theta_{est,i}}\right)\left(\theta_{obs,i} - \overline{\theta_{obs,i}}\right)}{\sqrt{\sum_{i=1}^{n}\left(\theta_{est,i} - \overline{\theta_{est,i}}\right)^2}\sqrt{\sum_{i=1}^{n}\left(\theta_{obs,i} - \overline{\theta_{obs,i}}\right)^2}}$ | 1 |
| Mean Bias Error (MBE) | m³ m⁻³ | $Bias = \dfrac{\sum_{i=1}^{n}\left(\theta_{est,i} - \theta_{obs,i}\right)}{n}$ | 0 |
| Root Mean Square Error (RMSE) | m³ m⁻³ | $RMSE = \sqrt{\dfrac{\sum_{i=1}^{n}\left(\theta_{est,i} - \theta_{obs,i}\right)^2}{n}}$ | 0 |
| unbiased Root Mean Square Error (ubRMSE) | m³ m⁻³ | ubRMSE $= \sqrt{\dfrac{\sum_{i=1}^{n}\left(\left(\theta_{est,i} - \overline{\theta_{est,i}}\right) - \left(\theta_{obs,i} - \overline{\theta_{obs,i}}\right)\right)^2}{}}$ | 0 |
| Probability of Detection (POD) | - | $POD = \dfrac{H}{H + M}$ | 1 |
| False Alarm Ratio (FAR) | - | $FAR = \dfrac{F}{H + F}$ | 0 |
| Critical Success Index (CSI) | - | $CSI = \dfrac{H}{H + M + F}$ | 1 |

Note: n is the observations number (1827) of each in situ station (58 stations in total). $\theta_{est,i}$ and $\theta_{obs,i}$ are model-derived
RZSM products and in situ measurements (m³ m⁻³), respectively ; $\overline{\theta_{est,i}}$ and $\overline{\theta_{obs,i}}$ are the mean of $\theta_{est,i}$ and $\theta_{obs,i}$ across the
entire research period; H is the number of rainfall events that are recognized by model and in-situ measurements; M is the
number of measured rainfall events that are not recognized by model product; F is the number of model-based rainfall events
that are not recognized by in situ measurements.
**3.3 Seasonal anomaly**

Soil moisture products may exhibit large differences across timescales (e.g., sub-seasonal, mean seasonal and

inter-annual) (Draper and Reichle, 2015, Gruber et al., 2020). In order to avoid seasonal effects, the soil moisture
products are commonly decomposed into different frequency components (e.g., the raw soil moisture and monthly
soil moisture anomaly). In this study, monthly anomaly time-series of root-zone soil moisture are calculated based
on the moving-average decomposition method. The difference to the mean is divided by the standard deviation
(stdev) for a moving-average window of five weeks (Rüdiger et al., 2009, Albergel et al., 2012). The moving
window F is defined as follow, for each RZSM estimate or observation at day (t), F=[t-17:t+17]. If there are at





least five measurements available in this period, the moving-average value and standard deviation of root-zone
soil moisture are calculated. The anomaly is given as following equation:
$RZSM_{anomaly}(t) = \frac{RZSM(t) - \overline{RZSM(F)}}{stdev(RZSM(F))}$                                    (3)
where $RZSM(t)$ and $RZSM_{anomaly}(t)$ denote raw RZSM and seasonal anomaly of RZSM at day $t$, respectively.
Equation (3) is applied to model-derived and in situ RZSM for comparison.





## 4 Results

### 4.1 Comparison between model-derived and in situ RZSM

Figure 2 shows time series and scatterplots of stations-averaged model-derived RZSM products (ERA-5, MERRA2, NCEP CFSv2, GLDAS_CLSM, CLDAS_NOAH, CLDAS, SMAP L4, SMOS L4) against the in situ measurements over the HRB, from 1 April 2015 to 31 March 2020. Generally speaking, all RZSM products capture the rapid temporal variations of in situ soil moisture observations, except for SMOS L4, which shows less rapid changes (left panel of Fig. 2). The in situ soil moisture exhibits a variation that ranges from 0.1 to 0.4 $m^3\,m^{-3}$. The range of NCEP CFSv2 and SMAP L4 RZSM is similar to the observed RZSM range (Fig. 2a and 2e). ERA5 and CLDAS present larger RZSM values, ranging from 0.2 to 0.5 $m^3\,m^{-3}$ (Fig. 2a and 2c). MERRA-2, GLDAS_CLSM and GLDAS_NOAH RZSM values range from 0.2 to 0.4 $m^3\,m^{-3}$ (Fig. 2a and 2c). This is a smaller interval than for the other products. SMOS L4 displays the smallest RZSM values, ranging from 0.1 to 0.3 $m^3\,m^{-3}$ (Fig. 2e). The right panel of Fig. 2 demonstrates the marked overestimation of in situ observations by ERA5 and CLDAS, and the underestimation by SMOS L4. In terms of correlation and ubRMSE, GLDAS_CLSM (R = 0.69, ubRMSE = 0.018 $m^3\,m^{-3}$, respectively) outperforms the other RZSM products while SMAP L4 presents the lowest RMSE and the lowest bias (0.03 and 0.04 $m^3\,m^{-3}$, respectively). SMOS L4 presents the worst performance in terms of correlation with R = 0.35.

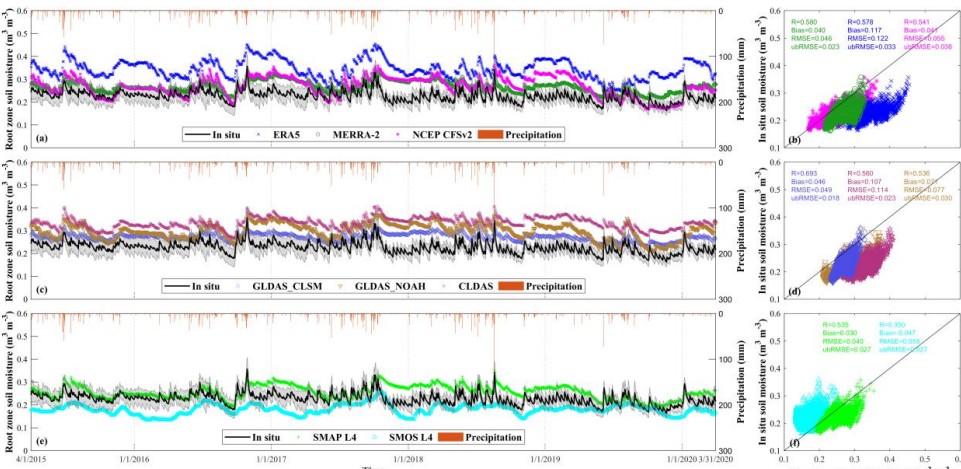

**Fig. 2 Stations-averaged RZSM (0-100 cm) comparison between model-derived RZSM and in situ soil moisture observations spanning the period from April 1, 2015 to March 31, 2020, including the time series (left panel) and scatterplots (right panel). The gray-shaded areas in the left panel represent the standard deviation of in situ stations observations within the HRB.**





Figure 3 shows the statistical distribution of the scores of the eight RZSM products across all in situ stations

in the HRB for three time periods of the seasonal cycle: the full annual cycle, the wet season from June to

September, and the dry season from October to May. The median and standard deviation values of the scores are

listed in Table 3. For the full annual cycle, the SMOS L4 RZSM presents a negative median bias of -0.050 $m^3$ $m^{-3}$

(equivalent to a soil moisture deficit of 50 kg $m^{-2}$) compared with the in situ measurements. All the other

products overestimate RZSM, from 0.033 $m^3 m^{-3}$ to 0.117 $m^3 m^{-3}$ (SMAP L4 and ERA5, respectively). All

temporal series of RZSM products correlate to the in situ measurements and correspond well to the precipitation

events. However, SMOS L4 time series are smoother than the observations and present the smaller correlation

(R = 0.21). The best correlation is obtained by GLDAS_CLSM (R = 0.50). This product also presents the

smallest ubRMSE value: 0.031 $m^3 m^{-3}$ against 0.048 $m^3 m^{-3}$ for SMOS L4. The reanalysis RZSM products

(ERA5, MERRA-2, NCEP CFSv2) tend to overestimate the in situ measurements. Among the three products,

MERRA-2 performs better with better average R and ubRMSE values (0.43 and 0.036 $m^3 m^{-3}$, respectively) than

ERA5 (R = 0.40, ubRMSE = 0.045 $m^3 m^{-3}$) and NCEP CFSv2 (R = 0.39, ubRMSE = 0.048 $m^3 m^{-3}$). ERA5

presents a large bias of 0.104 $m^3 m^{-3}$. The GLDAS_NOAH, GLDAS_CLSM, CLDAS and SMAP L4 products

also show an overestimation. GLDAS_CLSM outperforms CLDAS, GLDAS_NOAH and SMAP L4 with a

higher R value of 0.50 and a lower ubRMSE of 0.031 $m^3 m^{-3}$, followed by CLDAS (R = 0.44, ubRMSE = 0.035

$m^3 m^{-3}$), SMAP L4 (R = 0.37, ubRMSE = 0.039 $m^3 m^{-3}$) and GLDAS_NOAH (R = 0.35, ubRMSE = 0.043 $m^3 m^{-3}$

). CLDAS shows the largest wet bias value (0.116 $m^3 m^{-3}$) followed by ERA5 (0.104 $m^3 m^{-3}$). Because of the

large bias, CLDAS and ERA5 display the largest RMSE values (0.113 and 0.122 $m^3 m^{-3}$, respectively) among all

the RZMS products. SMAP L4 (R = 0.37, ubRMSE = 0.039 $m^3 m^{-3}$) performs better than SMOS L4 (R = 0.21,

ubRMSE = 0.048 $m^3 m^{-3}$). Overall, GLDAS_CLSM performs best among the eight RZSM products in terms of

R, ubRMSE and bias, followed by MERRA-2, CLDAS, SMAP, ERA5, NCEP CFSv2, GLDAS_NOAH, SMOS

L4. SMAP L4 presents the smallest bias.

It can be seen that the score values vary considerably across single stations in Fig. 3. In terms of correlation,

ERA5, MERRA-2, NCEP CFSv2 and GLDAS_NOAH all show their best R values varying from 0.59 to 0.67 over

the Xianghongdiankuxia station (number: 50701303) and SMAP L4 has its highest R value of 0.62 over the

Guanting station (number: 5042471). Both stations are located in the south of HRB where precipitation events are

more frequent. GLDAS_CLSM, CLDAS and SMOS L4 show their highest R values (0.67, 0.66 and 0.53,

respectively) over the Dahu, Youhe, and Baoji stations (numbers: 50701303, 50830439, and 50924801,

respectively), all of them located in the center of the HRB. In terms of bias, ERA5, MERRA-2, NCEP CFSv2,



GLDAS_NOAH, GLDAS_CLSM and CLDAS present smaller values in the north of HRB than in the south.
However, SMOS L4 has its smallest bias values in the south of HRB.

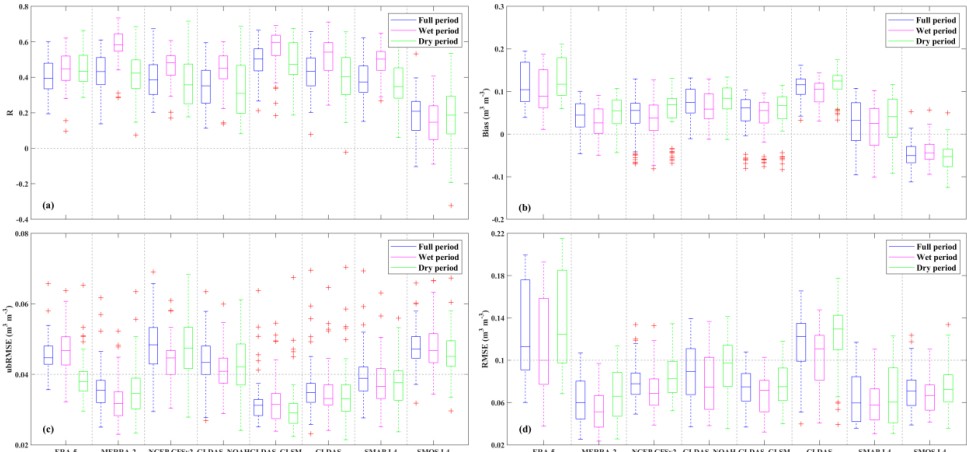


**Fig. 3 Single-station RZSM comparison between model-derived RZSM and in situ soil moisture observations for**
**different periods, including the Full period (from 1 April 2015 to 31 March 2020), Wet period (from June to**
**September) and Dry period (from October to May). Each outlier "+" represents an in situ station. The boxplot is**
**represented by the nonoutlier minimum $\left(Q1 - 1.5 \times (Q3 - Q1)\right)$, lower quartile Q1 (25th percentile), median Q2**
**(50th percentile), upper quartile Q3 (75th percentile), nonoutlier maximum $\left(Q3 + 1.5 \times (Q3 - Q1)\right)$, respectively.**
In order to eliminate the seasonal effects and to investigate the capacity of the products to represent the day-
to-day variability of RZSM, a moving-average window of five weeks is used to calculate the monthly anomaly
time-series of RZSM. Figure 4 displays a comparison of the scores on soil moisture anomalies. It can be seen
that statistical metrics based on in situ validation for monthly anomaly time-series of RZSM generally display
similar trends to that of in situ validation for raw RZSM time-series in terms of R and ubRMSE. However, some
differences can be observed. Anomaly R values are larger than raw R values for ERA5, MERRA-2, NCEP
CFSv2, CLDAS and SMAP L4 products. On the other hand, GLDAS_NOAH, GLDAS_CLSM and SMOS L4
products present lower anomaly R values than raw R values (Table 3). In general, the overall performance of the



eight RZSM products is better during the wet season than for the full annual cycle and the dry season.

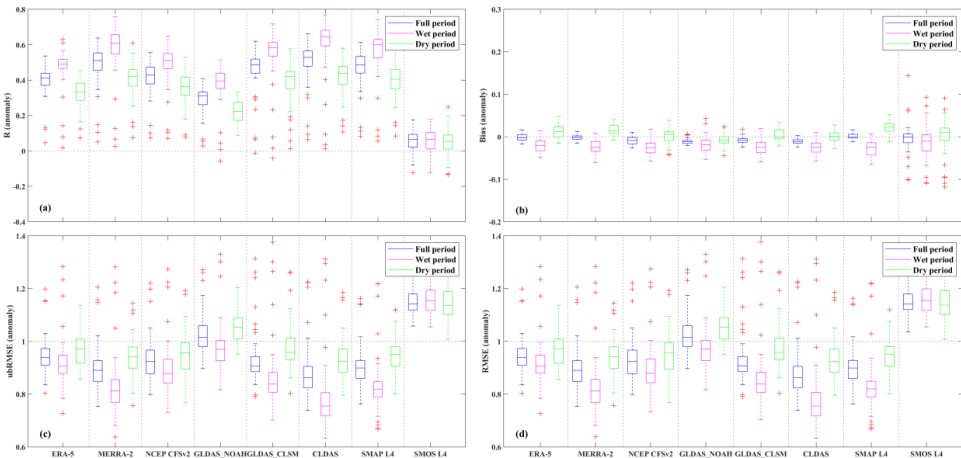


**Fig. 4 Same as Fig. 3, but for the monthly anomaly.**




Table 3 Statistical metrics of eight RZSM products validated by in-situ measurements from April 1, 2015 to March 31, 2020: Median (Std).

| Dataset | Soil Layer | Period | In situ validation (raw) | | | In situ validation (anomaly) | | |
|---|---|---|---|---|---|---|---|---|
| | (cm) | | R | ubRMSE | Bias | R | ubRMSE | Bias (anomaly) |
| ERA-5 | 0-100 | Full | 0.40 (0.10) | 0.045 (0.005) | 0.104 | 0.41 (0.08) | 0.94 (0.07) | -0.00 (0.01) |
| | | Wet | 0.45 (0.10) | 0.047 (0.006) | 0.089 | 0.49 (0.11) | 0.91 (0.09) | -0.02 (0.02) |
| | | Dry | 0.43 (0.10) | 0.038 (0.006) | 0.117 | 0.33 (0.08) | 0.97 (0.06) | 0.01 (0.01) |
| MERRA-2 | 0-100 | Full | 0.43 (0.10) | 0.036 (0.007) | 0.044 | 0.51(0.11) | 0.89 (0.09) | -0.00 (0.01) |
| | | Wet | 0.58 (0.09) | 0.032 (0.006) | 0.026 | 0.61 (0.14) | 0.81 (0.12) | -0.03 (0.02) |
| | | Dry | 0.42 (0.12) | 0.035 (0.008) | 0.055 | 0.42 (0.10) | 0.94 (0.07) | 0.02 (0.01) |
| NCEP CFSv2 | 0-100 | Full | 0.39 (0.11) | 0.048 (0.008) | 0.056 | 0.43 (0.10) | 0.92(0.08) | -0.01 (0.01) |
| | | Wet | 0.48 (0.09) | 0.045 (0.006) | 0.038 | 0.51 (0.12) | 0.88 (0.10) | -0.03 (0.02) |
| | | Dry | 0.36 (0.14) | 0.047 (0.010) | 0.069 | 0.36 (0.09) | 0.96 (0.08) | 0.01 (0.02) |
| GLDAS_NOAH | 0-100 | Full | 0.35 (0.12) | 0.043 (0.007) | 0.075 | 0.31 (0.08) | 1.02 (0.07) | -0.01 (0.01) |
| | | Wet | 0.45 (0.11) | 0.041 (0.006) | 0.059 | 0.40 (0.11) | 0.97 (0.11) | -0.02 (0.02) |
| | | Dry | 0.31 (0.15) | 0.042 (0.008) | 0.084 | 0.22 (0.06) | 1.05 (0.06) | -0.01 (0.01) |
| GLDAS_CLSM | 0-100 | Full | **0.50** (0.09) | **0.031** (0.007) | 0.061 | 0.49 (0.12) | 0.91 (0.10) | -0.01 (0.01) |
| | | Wet | **0.60** (0.11) | **0.031** (0.007) | 0.055 | 0.58 (0.15) | 0.84 (0.13) | -0.03 (0.02) |
| | | Dry | **0.47** (0.12) | **0.029** (0.007) | 0.067 | 0.42 (0.11) | 0.96 (0.086) | **0.00** (0.01) |
| CLDAS | 0-100 | Full | 0.44 (0.12) | 0.035 (0.008) | 0.116 | **0.53** (0.12) | **0.862** (0.10) | -0.01 (0.01) |
| | | Wet | 0.54 (0.11) | 0.033 (0.007) | 0.105 | **0.65** (0.16) | **0.76** (0.14) | -0.02 (0.02) |
| | | Dry | 0.40 (0.14) | 0.033 (0.009) | 0.125 | **0.44** (0.10) | **0.93** (0.08) | 0.00 (0.01) |
| SMAP L4 | 0-100 | Full | 0.37 (0.10) | 0.039 (0.007) | **0.033** | 0.49 (0.11) | 0.90 (0.08) | **0.00** (0.01) |
| | | Wet | 0.50 (0.08) | 0.037 (0.007) | **0.025** | 0.60 (0.14) | 0.81 (0.11) | -0.02 (0.02) |
| | | Dry | 0.35 (0.12) | 0.038 (0.008) | **0.041** | 0.41 (0.09) | 0.95 (0.07) | 0.02 (0.01) |
| SMOS L4 | 0-100 | Full | 0.21 (0.13) | 0.048 (0.007) | -0.050 | 0.06 (0.06) | 1.14 (0.05) | -0.00 (0.03) |
| | | Wet | 0.15 (0.13) | 0.047 (0.007) | -0.045 | 0.07 (0.07) | 1.16 (0.06) | **-0.01** (0.05) |
| | | Dry | 0.19 (0.16) | 0.045 (0.007) | -0.053 | 0.05 (0.08) | 1.14 (0.06) | 0.01 (0.04) |

Note: Bold values denote the optimal values for each period (full, wet and dry periods). (Std) denotes the standard deviation.

**4.2 Intercomparison of eight RZSM products**

Figure 5 displays the comparison in pairs of the eight RZSM products for grid cells located over the in situ stations. Overall, all RZSM products show good consistency, except for SMOS L4. The correlation coefficient R with any of the seven other RZSM products varies from 0.30 (MERRA-2 vs. SMOS L4) to 0.95 (SMAP L4 vs. MERRA-2), with an average value of 0.71. The mean bias varies from -0.067 $m^3\,m^{-3}$ (MERRA-2 minus CLDAS) to 0.165 $m^3\,m^{-3}$ (ERA5 minus SMOS L4) with an average value of 0.037 $m^3\,m^{-3}$. The ubRMSE varies from 0.010 $m^3\,m^{-3}$ (MERRA-2 vs. SMAP L4) to 0.040 $m^3\,m^{-3}$ (NCEP CFSv2 vs. SMOS L4) with an average value of 0.024 $m^3\,m^{-3}$. SMOS L4 differs most from the other products. The correlation coefficient R between SMOS L4 and the other seven RZSM products varies from 0.30 (MERRA-2 vs. SMOS L4) to 0.41 (GLDAS_NOAH vs. SMOS L4) with an average value of 0.35, and the mean bias varies from 0.077 $m^3\,m^{-3}$ (SMAP L4 minus SMOS L4) to 0.165 $m^3\,m^{-3}$ (ERA5 minus SMOS L4) with an average value of 0.112 $m^3\,m^{-3}$. The ubRMSE varies from 0.023 $m^3\,m^{-3}$ (GLDAS_CLSM versus SMOS L4) to 0.400 $m^3\,m^{-3}$ (NCEP CFSv2 vs. SMOS L4) with an average value of 0.031 $m^3\,m^{-3}$.

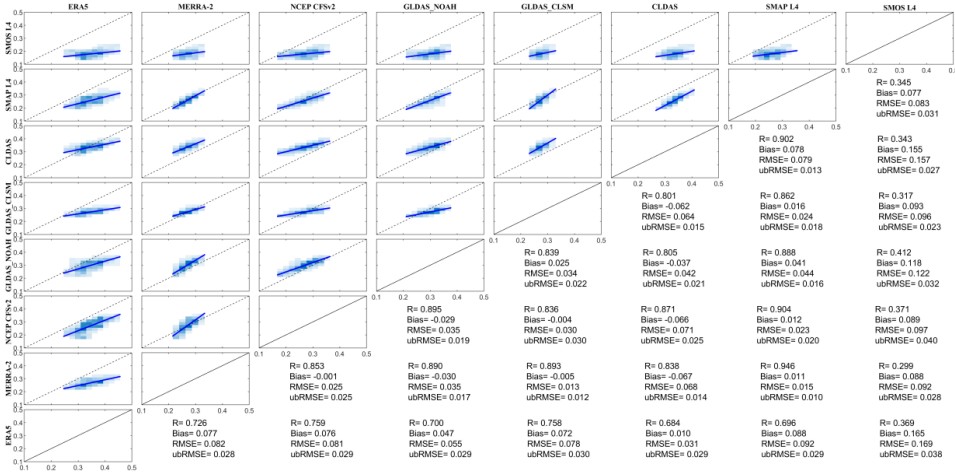

**Fig. 5 Comparison of different RZSM products (volumetric water content, $m^3\,m^{-3}$) with each other. The scatterplots and their corresponding statistics are located on opposite sides of each other, that is, the scatterplot of the data pair SMOS L4-ERA5 is in the top left-hand corner, while the respective statistical values are found in the bottom right-hand corner. Darker regions show a higher density of data point.**

Figure 6 shows the histograms of normalized RZSM of the eight model-derived products and of in situ observations. The relative frequency distribution corresponded to normalized soil moisture interval varies considerably across different RZSM datasets. All soil moisture datasets are almost normally distributed with one





clear peak. However, the observed RZSM distribution is skewed towards low values and the most frequent

normalized RZSM class ranges between 0.3 and 0.4. The MERRA-2, GLDAS_CLSM, SMAP L4, and ERA5

products display the same behavior. On the other hand, SMOS L4, NCEP CFSv2 and CLDAS have a relative

frequency peaking at a range of 0.4-0.5. GLDAS_NOAH even peaks at 0.5-0.6, and is clearly skewed toward the

wet end.

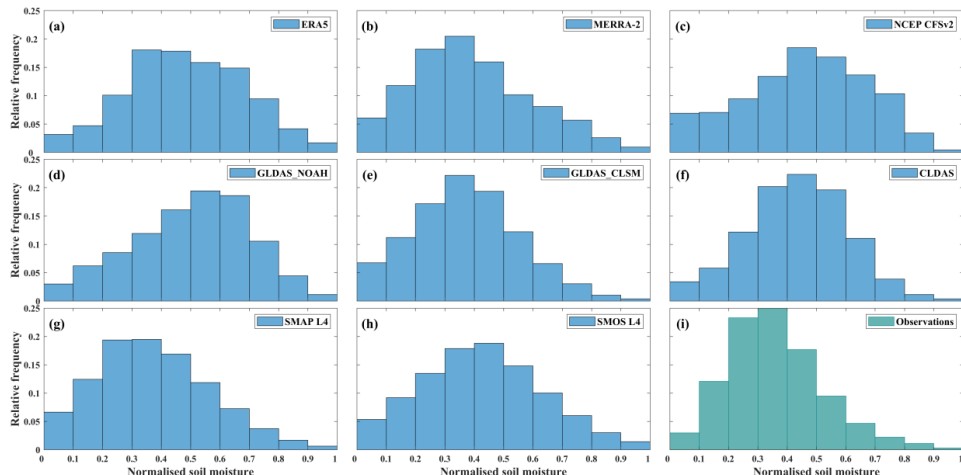

**Fig. 6 Histograms showing the relative frequency (vertical axis) of the various normalized RZSM datasets and in situ observations.**



## 381    5 Discussions

### 382    5.1 What is the impact of uncertainties of meteorological forcing data?

The meteorological forcing considered as one of the most important and direct factors influences the accuracy
of LSM simulations, especially precipitation and air temperature (Reichle et al., 2012, Yang et al., 2020, Zeng et
al., 2021). Precipitation and air temperature global forcing data are used in the generation of all RZSM products
except for SMOS L4. These forcing data were compared with reference data derived from in situ observations,
extracted from the China ground rainfall and air temperature gridded dataset. Figure 7 and Figure 8 show the
difference between global and ground-based precipitation. A daily precipitation amount less than 1 mm is
considered as a no-rain criterion. During the period from 1 April 2015 to 31 March 2020, the mean yearly
precipitation amount of global products (SMAP: 1024 mm yr$^{-1}$, GLDAS_NOAH: 988 mm yr$^{-1}$, GLDAS_CLSM:
986 mm yr$^{-1}$, MERRA-2: 974 mm yr$^{-1}$, NCEP CFSv2: 951 mm yr$^{-1}$, ERA5: 880 mm yr$^{-1}$) overestimates the ground-
based observations (840 mm yr$^{-1}$) by 22, 18, 17, 16, 13, and 5 %, respectively. In addition, the mean frequency of
rainy days (131, 114, 114, 113, 114, 126 d yr$^{-1}$) is larger than observed (97 d yr$^{-1}$) due to the drizzle effect often
produced by AGCM (Piani et al., 2010, Velasquez et al., 2020). For precipitation events exceeding a daily
precipitation amount of 50 mm d$^{-1}$, the global precipitation products tend to underestimate the in situ precipitation
observations (Fig. 7).

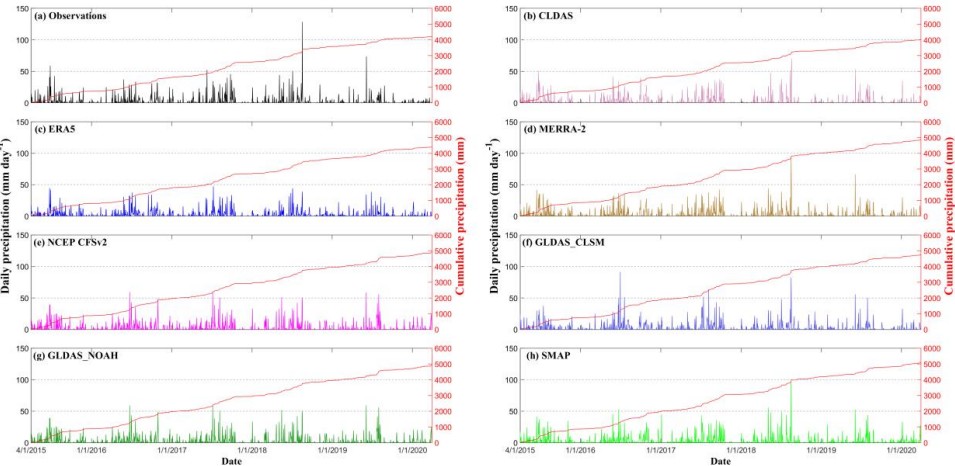


**Fig. 7 Stations-averaged daily precipitation and cumulative precipitation time series comparison between model-**
**derived precipitation and in situ precipitation observations.**



The larger precipitation amount and frequency could be a reason of the overestimation of soil water storage
by RZSM products generated by LSMs. We also quantitatively evaluated the model-derived precipitation by
comparing them with ground-based precipitation, to investigate the impacts of precipitation accuracy on the
performance of RZSM products (Fig. 8). It can be seen that, overall, the precipitation products are consistent with
observed precipitation, with R values generally above 0.4 (left panel of Fig. 8). MERRA-2, ERA5,
GLDAS_CLSM, SMAP L4, and ERA5 show strong precipitation detection ability with POD value above 0.6 (the
right panel of Fig. 8). The R value between model-derived and ground-based precipitation is not directly related
to the POD value. For example, NCEP CFSv2 does not perform as well as ERA5 in terms of POD but presents
better R values. In terms of R, RMSE, CSI, POD and FAR, the precipitation of MERRA-2 and GLDAS_CLSM
performs best among all products. This may explain the relatively better agreement of MERRA-2 and
GLDAS_CLSM RZSM with in situ data in terms of anomaly correlation (Fig. 4). For most reanalysis products,
the precipitation used to drive different LSMs was generated by AGCMs through the assimilation of atmospheric
temperature, humility and wind observations (Reichle et al., 2017). In addition, MERRA-2 model-generated
precipitation was corrected with two gauge-based precipitation observations before driving the land surface water
budget: (1) the NOAA CPCU gauge-based analysis of global daily precipitation product at 0.5° spatial resolution
and (2) the CMAP precipitation product based on merging gauge-based observations with satellite-derived
estimates at 2.5° spatial resolution. The MERRA-2 model-generated precipitation correction was implemented in
the coupled land-atmosphere reanalysis system, which may contribute to the high consistency with the ground-
based precipitation.



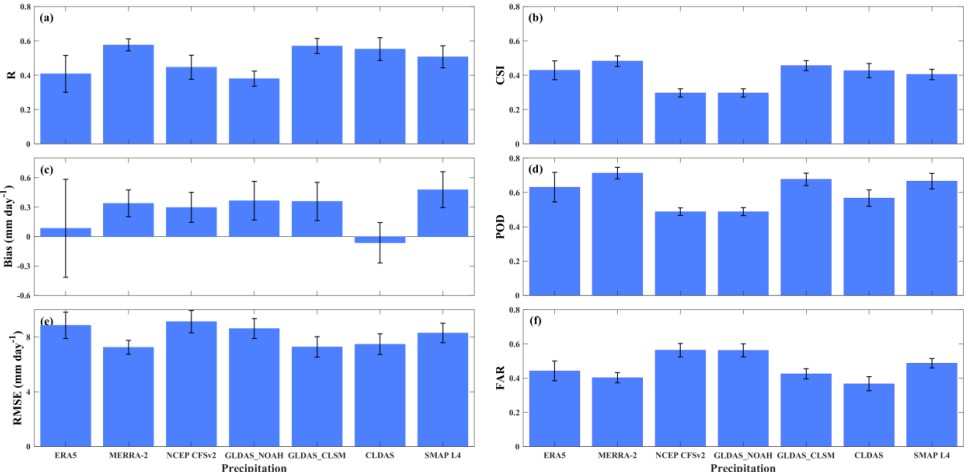


**Fig. 8 Summary of error metrics of model-derived precipitation data against in situ precipitation observations (left**

**panel), right panel shows the detection ability of model-derived precipitation to reproduce the observed precipitation.**

**The blue histogram represents the median and black error bar represents the standard deviation.**

Unlike the global products mentioned above, CLDAS (806 mm yr$^{-1}$) underestimates the yearly precipitation
amount by 13 %, and the precipitation frequency (99 days yr$^{-1}$) is close to the ground-based observation. Hence,
the CLDAS multi-LSMs should have produced smaller RZSM values being driven by CLDAS precipitation than
by the ground-based precipitation, but the CLDAS RZSM product overestimates the in situ observations by
0.116 m$^3$ m$^{-3}$ (Table 3). Therefore, precipitation may be not the dominant factor for the overestimation of RZSM
for CLDAS (Bi et al., 2016, Qin et al., 2017). Apart from precipitation, the performance of model-generated
RZSM products was also affected by uncertainties on air temperature, soil properties, soil stratification, model
parameterizations, etc.
Air temperature is another key factor after precipitation determining the accuracy of LSM simulations by
controlling soil evaporation and plant transpiration. In order to investigate the impacts of air temperature on the
performance of RZSM simulations, we evaluated the air temperature data derived from ERA5, MERRA-2, NCEP
CFSv2, GLDAS_CLSM, CLDAS, GLDAS_NOAH and SMAP L4 by comparing them with the in situ
observations of daily air temperature. Figure 9 shows the model air temperature captures the observed temporal
variation with R values above 0.96. However, all of them show underestimation with negative bias values ranging
from -4.0 to -5.2 K. This issue was illustrated in previous studies (Wang and Zeng, 2012, Yang et al., 2020).
Generally speaking, the lower air temperature used to generate RZSM products triggers less evapotranspiration,
and more soil water storage. This is consistent with the overestimation of in situ observations by LSM-based


RZSM products (Bi et al., 2016, Yang et al., 2020). Comparing with precipitation, air temperature has better overall
correlation with in situ observations.

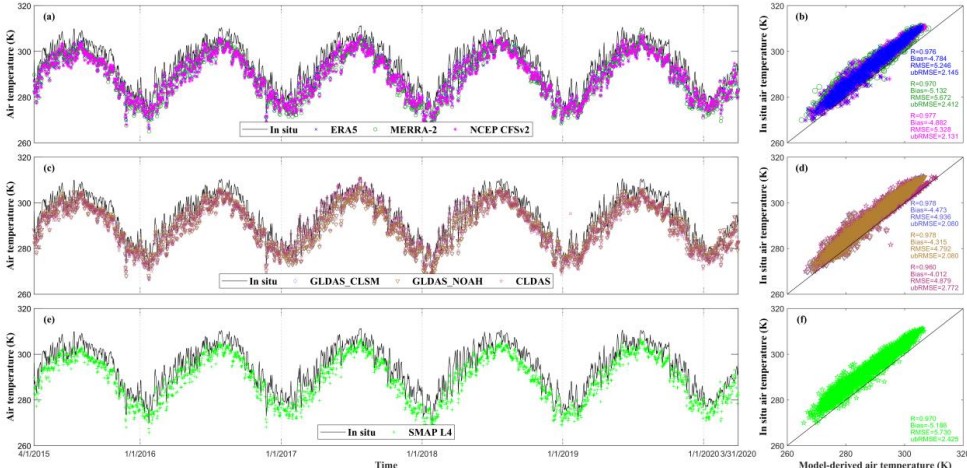


**Fig. 9 Same as Fig. 2, but for the air temperature.**

**5.2 Are soil properties correctly represented?**

Soil properties data (e.g., porosity) are key and time-invariant model parameter for LSM, because they

determine the physical structure of soil in the vadose zone, which controls the partition of precipitation into surface
runoff and infiltration. Previous studies have shown that FAO/UNESCO soil properties are affected by
uncertainties in different regions (Shangguan et al., 2013, Bi et al., 2016), Yang et al. (2020), (Xing et al., 2021,
Zheng et al., 2022). Here, four soil properties indicators, including clay and sand content, soil organic carbon
content and bulk density were chosen to investigate the difference among the FAO/UNESCO soil map of World,
HWSD and the reference soil data set developed by Shangguan et al. (2013). The soil properties data used in the
eight RZSM products are all derived from the FAO/UNESCO soil map of World except for CLDAS which used
the soil data developed by Shangguan et al. (2013). Figure 10 shows the reference dataset and HWSD generally
present similar characteristics, except for the slightly higher organic carbon content and lower sand content of the
reference dataset. Both of them differ from FAO/UNESCO soil properties data. FAO/UNESCO overestimates the
clay content for the top (0-30 cm) and subsurface (30-100 cm) soil layers. The sand content is also overestimated
for the subsurface layer but it is underestimated for the top layer. Generally speaking, the ability of soil to retain
water is related to the soil texture, because water molecules are more tightly attached to the soil particles of fine-
textured clay than coarse-textured sand. So, the clay has stronger water retention capacity and higher water content
stored in the soil than the sand at the same matric potential. In addition, the organic carbon content also influences

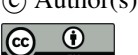

the water holding capacity of the soil. Commonly, high soil organic carbon content is related to high soil porosity
and to low bulk density. As a result, water can infiltrate more rapidly and more water flows through the soil and
can be held in the soil (Bot and Benites, 2005, Reichle et al., 2017). Moreover, increasing porosity may increase
the specific surface area of soil particles, which further increases the water holding capacity of the soil, and more
water content can be retained in the soil. Therefore, the inaccurate FAO/UNESCO soil properties data used in
LSMs can explain the overestimation of soil moisture by various RZSM products relative to the ground-based
observations. It is promising to observe that the accuracy of LSM-based RZSM can be improved using HWSD
rather FAO/UNESCO soil properties data.

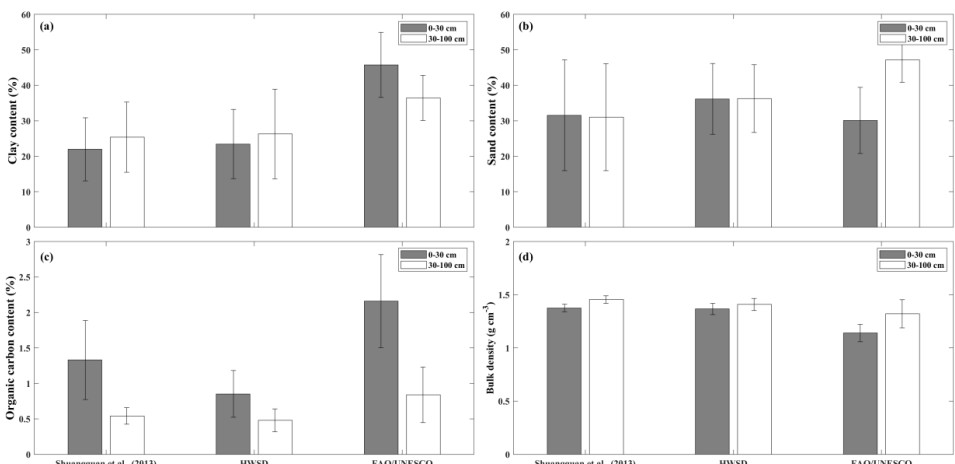


**Fig. 10 Soil properties data produced FAO used in (ERA5, MERRA2, NCEP CFSv2, GLDAS_NOAH, GLDAS_CLSM,**
**SMAP and SMOS), HWSD and reference soil properties data Shangguan et al. (2013) used in CLDAS. The histogram**
**(gray: 0-30 cm; white: 30-100 cm) represents the median and black error bar represents the standard deviation.**
Soil stratification may affect the accuracy of LSM-based RZSM through impeding the water transfer from
the surface layer to the root zone layer. The soil column in the Huaibei plain can basically be divided into three
layers, including the plough layer (0–16.6 cm), black soil layer (16.6–49.3 cm) and lime concretion layer (49.3–
138.3 cm). The discrepancy for soil properties data between the plough layer and black soil layer is higher than
that of black soil layer and lime concretion layer (see Fig. S1). The fine-textured clay content and coarse-textured
sand content of plough layer is obviously less and slighter higher than that of black soil layer, respectively. Due to
long-term human activity, the physicochemical characteristics of the soil plough layer has been changed
considerably. The agricultural activity (fertilization and plough) significantly increases the soil organic carbon
content and porosity of plough layer relative to the black soil layer and lime concretion layer. High porosity leads



to high hydraulic conductivity and infiltration capacity (Zha et al., 2015). Therefore, there exists a relative
impermeable interface due to the fact that the infiltration rate of plough layer is higher than that of black soil layer.
Under the circumstances, when the water content of upper soil layer reaches field capacity, the subsurface flow
emerges. As rainfall accumulates, the subsurface water may either flow in the horizontal direction or accumulate
in the vertical direction with weak lateral drainage condition and evaporate. These processes may be not well
represented by LSMs.

**5.3 Why are MERRA-2 and SMAP L4 RZSM highly correlated?**

The very good correlation and low ubRMSE between MERRA-2 and SMAP L4 shown in Fig. 5 may be
partly attributed to the fact that SMAP L4 and MERRA-2 share the same surface meteorological forcing generated
from GEOS-5. Moreover, the SMAP L4 precipitation data generated by NASA GEOS-5 is corrected with the
NOAA CPCU gauge-based analysis of global daily precipitation product. The MERRA-2 precipitation data are
also corrected with CPCU but the Climate Prediction Center Merged Analysis of Precipitation (CMAP) product
is used too. Since precipitation is the dominant driver of the land surface water cycle, this can explain the large R
value between SMAP L4 and MERRA-2 RZSM products. In addition, both SMAP L4 and MERRA-2 use the
CLSM.

**5.4 How do different LSMs parameterizations affect model-derived RZSM?**

The accuracy of model-generated RZSM may depend on uncertainties in model parameterizations (Reichle
and Koster, 2003). Regarding the water and energy balance represented in different LSMs, the partitioning of net
radiative energy into latent heat flux, sensible heat flux and ground heat fluxes, the partitioning of the precipitation
into interception, evaporation, infiltration and runoff as well as the transfer and exchange of water and heat in the
vadose zone vary considerably (Koster et al., 2000, Chen et al., 2013, Xia et al., 2014, Reichle et al., 2017). For
instance, NOAH LSM, HTESSEL and CLM have 4-, 4- and 10-layer vertical levels for soil moisture and
temperature, respectively (Oleson et al., 2004, Rui et al., 2021). CLSM represents vertical levels for soil moisture
in surface layer (0-2 cm) and root zone layer (0-100 cm) but has six layers for soil temperature (Rui et al., 2021).
The computational unit in CLSM is hydrological catchment, and the adjacent catchments have no fluxes exchange
such as groundwater or runoff (Koster et al., 2000, Reichle and Koster, 2003). The computational unit in CLM is
grid cell, where the spatial heterogeneity of land surface is represented by three nested subgrid hierarchy (Oleson
et al., 2004). NOAH LSM describes the incomplete hydrological cycle process at the grid scale, and it neglects the
heterogeneity of soil, which has great effect on infiltration and the generation and convergence of runoff (Wang



and Chen, 2013). HTESSEL also calculates the water and energy balance at the grid scale and neglects lateral
exchange of soil water between adjacent grid cells. Regarding the surface runoff parameterizations, CLM adopts
a conceptual form of the original TOPMODEL to configure the runoff parameters. The surface runoff is calculated
through saturated and unsaturated fractions combined with the sum of the melt water from snowpack and liquid
precipitation falling to the land surface (Oleson et al., 2004). A Simple Water Balance (SWB) model is used to
parameterize surface runoff obtained from precipitation minus the maximum infiltration in the NOAH LSM, and
the process of runoff generation is considered only in the vertical direction. HTESSEL also adopts the SWB model
to calculate surface runoff with an additional snowmelt item, but different maximum infiltration schemes were
adopted in HTESSEL and NOAH LSM, respectively. CLSM accounts for topography on the spatial variability of
soil water and its effect on evaporation and runoff into account using TOPMODEL. In each catchment, CLSM
incorporates different parameterization schemes describing the energy budget processes in specific hydrological
regimes into each hydrological catchment model depicting the redistribution of water based on topography, which
results in reliable estimates of evaporation and runoff (Ducharne et al., 2000, Koster et al., 2000)In fact, the range
of runoff generation area changes in the horizontal direction when precipitation occurs (Wang et al., 2016).
Therefore, the different parameterizations of infiltration and runoff generation lead to the differences in model-
derived RZSM products.

### 5.5 How does the mismatch of spatial scale affect the evaluation results?

Except for the model- and the observation-generated soil moisture errors, the mismatch of spatial scale
between grid-scale soil moisture simulations and point-scale observations also introduces additional errors. As the
statistical metrics shown in section 4.1, it can be seen that the R and ubRMSE between regionally-averaged RZSM
products and stations-averaged in situ observations overall outperforms that between RZSM grid value and point-
scale observations at each in situ station located within the model grid cell. For the latter, grid-based RZSM has
poor representativeness of soil moisture within a grid cell exhibiting high spatial variability due to the effect of
different characteristics of underlying surface and meteorological forcing. The latter comparison will introduce the
representativeness error (Xia et al., 2014, Bi et al., 2016). By contrast, the former comparison improves the
representativeness of the grid-based RZSM and reduces the spatial noise (Wang and Zeng, 2012, Xia et al., 2014,
Bi et al., 2016, Zheng et al., 2022). Moreover, it is promising to reduce the uncertainty of spatial resampling by
upscaling the sparse ground-based observations match to the footprint-scale satellite soil moisture retrievals or
model grid scale through time stability concepts, block kriging, field campaign data or LSM and further improve
the reliability of soil moisture validation (Crow et al., 2012).



### 5.6 Why is SMOS L4 RZSM underestimated?

The SMOS L4 RZSM was obtained through SMOS L3 3-day SSM combined with modified exponential filter (Pablos et al., 2018). Figure 11 shows the comparison of SMOS L3 SSM and L4 RZSM against the in situ soil moisture observations. It can be observed that that both SMOS L3 SSM and L4 RZSM are smaller than the in situ observations with average bias value of -0.069 and -0.047 $m^3 m^{-3}$, respectively. Meanwhile, a previous study (Ford et al., 2014) has pointed that the error between in situ observations and estimation is far more than the error caused by the exponential filter model by partitioning the total error composed of the exponential filter model and inherent SMOS in situ differences. The underestimation of in situ observations by SMOS L3 SSM has been reported in previous studies (Djamai et al., 2015, Cui et al., 2017, Pablos et al., 2018, Ma et al., 2019, Wang et al., 2021). Therefore, it can be inferred that the underestimation of in situ observations by SMOS L3 SSM propagates to SMOS L4 RZSM. The microwave signal at L-Band is sensitive to soil moisture, to soil temperature and to the Vegetation Optical Depth (VOD) (Kerr et al., 2012). Using the L-band Microwave Emission of the Biosphere (L-MEB) model (Wigneron et al., 2007), SMOS L3 soil moisture and Vegetation Optical Depth (VOD) can be simultaneously retrieved using multi-angular (~0-60°) and dual-polarization TB measurements from several orbits (Al Bitar et al., 2017). Soil temperature, VOD, SSM and soil roughness are the most sensitive parameters in the radiative transfer model (Wang et al., 2016, Fernandez-Moran et al., 2017). Among the four variables, VOD and soil temperature are often used to investigate the accuracy of SMOS L3 soil moisture retrievals (Cui et al., 2017, Wang et al., 2021, Zheng et al., 2022). Figure S2 shows that the model-generated soil temperature captures the temporal variation of the ground-based observations very well with R values above 0.97 except for NCEP CFSv2 and SMOS L4 R values smaller than 0.9. Except for CLDAS (bias = 1.3 K), all model-generated temperature products show an underestimation with a mean bias value ranging from -9.8 to -1.9 K. The SMOS L4 RZSM is derived from SMOS L3 SSM (descending orbit, 06:00 PM), so the SMOS L3 soil temperature was compared with the in situ surface temperature observations at 06:00 PM and shows the negative bias value of -9.8 K, which is consistent with the conclusion drawn in previous studies (Cui et al., 2017, Ma et al., 2019, Wang et al., 2021, Zheng et al., 2022). In the SMOS L3 retrieval algorithm, underestimating soil temperature will cause the overestimation of soil emissivity, which finally may lead to the underestimation of soil moisture retrievals (Wang et al., 2021). VOD is also an important factor determining the accuracy of satellite-derived L4 soil moisture retrievals. In the study, the SMOS L3 SSM was found to be positively correlated with VOD with average R value of 0.28 (Fig. S3). Previous studies have illustrated that the VOD retrievals from SMOS may be noisy, which could be attributed to the effect of radio frequency interferences. Several authors showed that high VOD retrievals lead





to high soil moisture retrievals (Cui et al., 2017, Wang et al., 2021, Zheng et al., 2022). However, it cannot be
inferred whether the VOD retrievals from SMOS lead to the overestimation or underestimation of SMOS L3 SSM.

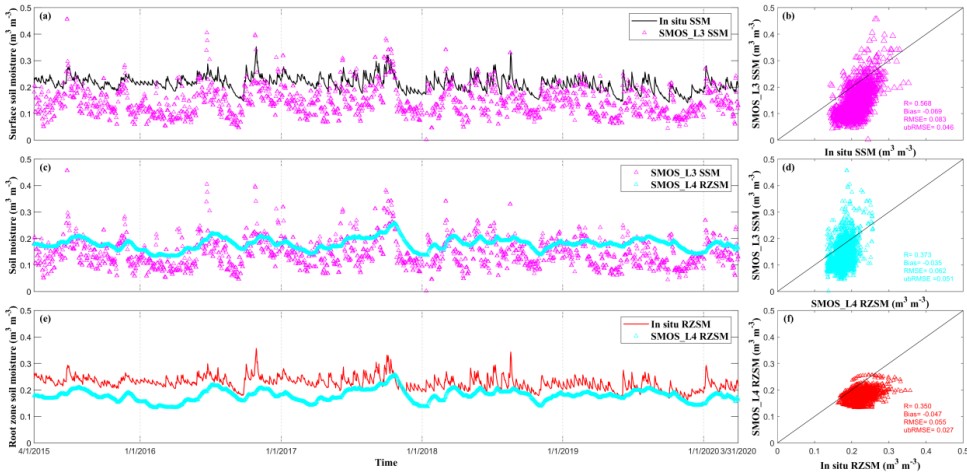


**Fig. 11 Comparison of time series (left panel) and scatterplots (right panel) of SMOS L3 SSM vs. in situ SSM (Fig. 11a**
**and b), SMOS L3 SSM vs. SMOS L4 RZSM (Fig. 11c and d) and SMOS L4 RZSM vs. in situ RZSM (Fig. 11e and f).**





**6 Conclusion**

In this study, eight RZSM products were quantitatively evaluated against observations from 58 in situ soil moisture stations over the HRB in China. Statistical metrics of R, mean bias, RMSE and ubRMSE were used to quantify the performance of different RZSM products. The impact of several potential perturbing factors on the uncertainty of model-derived RZSM products was investigated. These factors included meteorological forcing variables (precipitation and air temperature), soil properties (organic matter, clay and sand content), soil stratification, model parameterizations and spatial scale mismatch. The main conclusions drawn in this study were as follows:

(1) GLDAS_CLSM performed best among the RZSM products based on LSMs over the HRB in terms of R, ubRMSE and mean bias, followed by MERRA-2, CLDAS, SMAP, ERA5, NCEP CFSv2, and GLDAS_NOAH. The SMOS L4 product presented the lowest performance. All LSM-based products overestimated RZSM with median bias values ranging from 0.033 $m^3\,m^{-3}$ (SMAP L4) to 0.116 $m^3\,m^{-3}$ (CLDAS). On the other hand, SMOS L4 underestimated RZSM with a median bias value of -0.050 $m^3\,m^{-3}$. ERA5 and CLDAS showed the largest bias values of 0.104 $m^3\,m^{-3}$ and 0.116 $m^3\,m^{-3}$, respectively.

(2) The correlation coefficient R between any two of the seven LSM-based RZSM products varied from 0.68 (ERA5 vs. CLDAS) to 0.95 (SMAP L4 vs. MERRA-2). The higher R value between SMAP L4 and MERRA-2 RZSM was attributed to the fact that SMAP L4 and MERRA-2 are both based on CLSM and on the same surface meteorological forcing generated from the NASA GEOS-5 in which precipitation was corrected with the gauge-based CPCU precipitation product. SMOS L4 did not correlate well with the other seven RZSM products with R ranging from 0.30 (MERRA-2) to 0.41 (GLDAS_NOAH) and with a negative bias ranging from -0.165 $m^3\,m^{-3}$ (SMOS L4 minus ERA5) to -0.077 $m^3\,m^{-3}$ (SMOS L4 minus SMAP L4).

(3) Precipitation could be the most important factor determining the accuracy of LSM-based RZSM. Apart from CLDAS, the various precipitation datasets all show an overestimation of the total precipitation amount and precipitation frequency (excessive number of occurrences of drizzle events). This may explain the overestimation of the in situ soil moisture observations by various RZSM products but not for CLDAS. Air temperature used to drive LSMs presented a cold bias ranging from -4.0 K (CLDAS) to -5.19 K (SMAP L4), which tended to decrease evapotranspiration and increase RZSM.

(4) The underestimation of RZSM SMOS L4 can be related to the underestimation of SMOS L3 SSM.



*Data availability.* The datasets presented in this study can be obtained upon request to the corresponding author

*Author contributions.* EL, YHZ, JCC and HSL conceptualized the project. EL led the investigation, determined
the methodology and wrote the original draft of the paper. All the co-authors contributed to the review and editing
of the paper.

*Competing interests.* The authors declare that they have no conflict of interest.

*Disclaimer.* Publisher's note: Copernicus Publications remains neutral with regard to jurisdictional claims in
published maps and institutional affiliations.

*Acknowledgement.* We acknowledge the European Centre for Medium-Range Weather Forecasts (ECMWF),
Goddard Earth Sciences Data and Information Services Center (GES DISC), National Center for Atmospheric
Research (NCAR), China Meteorological Administration (CMA), National Snow & Ice Data Center (NSIDC) and
Centre Aval de Traitement des Données (CATDS) for providing data free of charge.

*Financial support.* This research was funded by National Key Research and Development Program (grant nos.
2019YFC1510504); National Natural Science Foundation of China (grant nos. 41830752, 42071033 and

41961134003).



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
