# Peer review of "Evaluation of model-derived root-zone soil moisture over the Huai river basin"

_Hydrology and Earth System Sciences, 2023_

## Referee Comment (RC2)

REVIEW

Manuscript ID:  hess-2023-33

Title:              Evaluation of model-derived root-zone soil moisture over the Huai river basin

Authors:         Liu, E., Y. Zhu, J.-C. Calvet, et al.

The paper evaluates 7 global root-zone soil moisture products and one regional product over the Huai river basin in China for the period from April 2015 to March 2020.  The products are evaluated against each other and against in situ measurements from 58 sensor profiles.  The differences in the products and their performance against in situ measurements is related to the skill of the products' forcing inputs (precipitation, air temperature), model structure, and model parameters (soil texture).

The authors find that the GLDAS_CLSM product performs best overall and has the highest R and lowest ubRMSE values.  Root-zone soil moisture is overestimated w.r.t. the in situ measurements in the 7 products based on land surface modeling, which is traced to an overestimation of the precipitation inputs, too much drizzle, underestimation of the air temperature inputs, and errors in the prescribed soil texture.

The paper addresses a topic of interest to HESS readers and presents some important results about the skill of several widely-used global root-zone soil moisture products and one regional product. Unfortunately, the paper falls short in several aspects, as described in the major comments below.  The paper *may* be suitable for publication in HESS after major revisions.  However, given the gaps and errors in the submitted manuscript, it is more realistic to REJECT the paper at this time and encourage resubmission later if the authors are willing and able to address the shortcomings.  This gives the authors a better chance to produce a revised manuscript of sufficient quality without being constrained by the customary 2-month period for major revisions.

Major comments:

1) There are clear errors in the authors' description of the product characteristics that impact the interpretation of the results.
Lines 26-27, also Lines 592-593: "[MERRA-2 and SMAP L4] are based on the same LSM and on the same surface meteorological forcing…".  While both products use the Catchment model, and both use atmospheric forcing data generated with GEOS, the model versions and the forcing data are not the same.  MERRA-2 uses an older version of the Catchment model than SMAP L4 (Reichle et al. 2017, 10.1175/JHM-D-17-0063.1; Reichle et al. 2019), including (but not limited to) the difference that soil parameters are based on FAO data in MERRA-2 and on HWSD in SMAP L4.  Re. surface meteorological forcing, MERRA-2 uses an older GEOS version 5.12 at 0.5-deg resolution, whereas the forcing data in SMAP L4 is from the GEOS weather analysis ("Forward Processing", or FP) system with evolving versions 5.17 to 5.25 and at 0.25-degree resolution.   In MERRA-2, the CPCU precipitation is used in its native climatology whereas in SMAP L4 the CPCU precipitation is rescaled to an independent reference climatology.

These differences are not discussed in the paper but are critical for the interpretation of the results, especially those re. errors in the products' soil parameters and the cause of the RZSM bias.   For example, Figure 8 shows that SMAP L4 and MERRA-2 precipitation metrics vs in situ measurements are different, but the authors simply ignore this result and keep repeating that SMAP L4 and MERRA-2 have the same forcing data (e.g., Line 592).  The fact that MERRA-2 also uses CMAP precipitation inputs is irrelevant (Line 493) because CMAP is used in MERRA-2 only over Africa and the ocean.

I outlined the above differences between MERRA-2 and SMAP L4 in detail because I am most familiar with these products.  I did not exhaustively review the accuracy of the other products' characteristics.  Descriptions of the other products may or may not contain additional errors.

2) Besides the errors in the product descriptions outlined above, there is insufficient information in the description of the products and analysis methods:
    a. Section 2.3 is very uneven across the subsections (individual product descriptions) in terms of the level of detail provided.  E.g., in some cases the horizontal resolution of the product is not provided.
    b. For MERRA-2 the native 0.5-deg data is interpolated to 0.25 degree, but there is no comparable statement for SMAP L4.  Are the SMAP L4 data used in their native (9-km) resolution?  Aggregated to 0.25 degree?
    c. ERA5 includes a soil moisture and screen-level temperature and humidity analysis.  This analysis clearly impacts the ERA5 soil moisture estimates but is not mentioned even once in the paper.
    d. Much of the text in the subsections of section 2.3 appears copied from the blurb of the descriptions on the products' websites.  In some cases, the text includes to the motivation of products' development from the preceding version. E.g., MERRA-2 is described in reference to the long-obsolete original MERRA data (Lines 159-163), and NCEP CFS v2.2 is described in reference to NCEP R1 and R2 (~Line 176).  The product descriptions in this paper should focus on each product's characteristics and how they differ from the other products examined here, not on how the products differ from older versions that are not examined here.
    e. The information on the specific air and soil temperature variables provided in the manuscript is not sufficient.  What are the air temperatures that are used in the comparison?  Are they at the 2-m screen level (T2m) or at the atmospheric model's lowest level?  The SMAP L4 product only provides the latter, which is used to drive the Catchment model.
    f. As for the simulated soil temperature, what is the layer depth for each product?  And what exactly is the in situ soil temperature shown in Fig 9?  The strong short-term variability of the in situ soil temperature (after averaging across 58 stations!) is highly suspect.  Could this be the surface (or skin) temperature?  The comparison here is almost certainly one of apples vs. oranges.
    g. What is the spatial resolution at which the temperature is validated?
    h. The in situ soil moisture measurements are taken at 8am local time (Line 109), but what about the in situ soil temperature measurements?  If the in situ soil temperature is taken at 8 am local time, are the soil temperatures from the products (Fig 9) also

sampled at 8am local time?  Are daily average soil temperatures from the products used?

    i.   It is not clear at what time step the second-order skill metrics (R, ubRMSE) are computed.  Are the metrics computed at daily time steps after aggregating all sub-daily products to daily time steps?

    j.   The in situ soil moisture data are QC'd (Lines 111-112) but there is no information on how many data points are actually used at each station.  The footnote of Table 2 suggests that 1827 (daily?) values are used at each of the 58 stations, but then no data would have been excluded by the QC process.  This is contradictory.

3) Many references are plainly incorrect or inappropriate, or are missing altogether:

    a.   Many of the references used for the products are not the first-hand references.  Rather, the references used are simply about applications of the data product.  This is not acceptable.  The authors must use first-hand references for the data products examined.  In other cases, inappropriate references are cited, or references are simply wrong.  Here are some of the problematic references (not meant to be a complete list):

        i.   Bi et al. 2016 is used in several places as a reference for GLDAS product characteristics but the paper is not a first-hand description of the GLDAS products.

        ii.   Line 51: The main ERA5 reference is Hersbach et al. QJRMS 2020.

        iii.   Line 46: Rienecker et al 2008 is not an appropriate reference for SMAP L4.

        iv.   Line 56: Reichle et al. 2017 is not an appropriate reference for NCEP CFSv2.

        v.   Line 80: Koster et al. 2016 is really McCarty et al. 2016.

        vi.   Lines 83-84: Need a reference for this statement about "layer stratification" (and also edit the statement for clarity); also in Lines 479-480.

    b.   There is no reference for the HWSD soil data!

    c.   References by the same lead author written in the same year are not distinguished properly.  E.g., there are two papers by Wang et al. 2016 and three papers by Reichle et al. 2017 in the list of references.  In the text, they are universally referred to as Wang et al. 2016 or Reichle et al. 2017, leaving the reader guessing which paper the authors refer to in each instance.

    d.   Data products such as SMAP L4 that have a digital object identifier must be cited with proper references that include the DOI.

4) The version of some of the products examined here are unclear.  E.g., what SMAP L4 version is used in the paper?  This has important implications on the interpretation of the results.  The precipitation forcing data in SMAP L4 versions 5 and 6 differs considerably, which is well documented in the products' validation reports (disseminated by NSDIC along with the data).

5) Like SMAP L4 and CLSM, the GLDAS_CLSM product also uses the Catchment model and precipitation forcing based on observations, but there are clear differences in the skill metrics between the three products.  One distinguishing feature of GLDAS_CLSM is the assimilation of GRACE TWS observations, which inherently contain information about root-zone soil moisture.  There is no discussion in the text on what the impact of the GRACE data assimilation may be on

the skill of GLDAS_CLSM vs. the other two Catchment-based products. (By the way, the Catchment model versions in GLDAS_CLSM and MERRA-2 are different but closer to each other than to the version used in SMAP L4.)

6) There is no reference for the in situ measurements other than that "data are available upon request". This should not be acceptable to HESS. For the study to be reproducible, the in situ measurements used here must be made readily accessible to interested researchers.

7) The discussion in section 5.4 is rambling and does not add useful information. All land models underpinning the products are "1-dimensional" models in the sense that grid cells (or computation elements) are not coupled horizontally other than through the forcing data. The authors make a mess of this simple fact by selectively pointing this out for some products (e.g., for CSLM in Line 506 and for HTESSEL in Line 511) but not others (e.g., Noah, Lines 509-510). The entire section is worthless and should be deleted.

Minor comments:

1) English style and grammar need some attention. Overall, the paper is readable but there are occasional English language errors.

2) The graphics generally lack clarity and quality.
    a. The axes labels (and other text within the graphics) are often too small to be readable (e.g., Figs 3 and 11)
    b. Color/symbols are sometimes difficult to distinguish (e.g., Figs 2 and 9)
    c. The resolution of the images embedded in the pdf is generally poor (e.g., Figs 2, 5, 7)

3) Inconsistent use of product names ("SMAP" vs "SMAP L4") (e.g., lines 322-323; Fig 7h label)

4) Line 337/Figure 3: Is the "nonoutlier minimum Q1 – 1.5 x (Q3 – Q1)" meant to reflect the "whiskers" of the box plot? If so, I don't see how this can be correct. Same for the "nonoutlier maximum" (Line 338)

5) Line 21: "The underestimated SMOS L3 SSM associated with low physical temperature triggers…" This is unclear. Do you mean "The underestimation of SMOS L3 SSM associated with…", or, perhaps more clearly, "The underestimation of the SMOS L3 SSM during cold conditions…" ???

6) Table 3 does not need a column for "Soil layer" because it is the same for each product.

7) Table 3 needs units for the bias. m3/m3 ??

8) Table 3: What is the "Bias (anomaly)" in the final column? Isn't this zero by construction?

9) Line 370: "histograms of normalized RZSM":  How was RZSM normalized?  I could not find this information.

10) Lines 101-103: The average annual precipitation in the Huai basin is listed as 888 mm precip and the average annual evaporation is listed as ranging from 900 to 1500 mm.  A relatively small fraction of the Huai basin is irrigated.  How can the average evaporation exceed precipitation by that much?

---

## Author Comment (AC1)

**Response to Reviewer #1's comments on the manuscript HESS-2023-33**

The authors thank the Reviewer #1 for her/his constructive and insightful comments that help us improve the quality of the manuscript. The original comments from Reviewer #1 are in black font, and our responses are in blue font.

**RC1: 'Comment on hess-2023-33', Anonymous Referee #1, 06 Mar 2023**

I thought it was an interesting study. I have one main comment: SMOS L4 is based on SMOS L3. From what I understand (Wigneron et al., 2021) ECMWF soil moisture data is used in the SMOS L3 retrieval algorithm.

Wigneron 2021, https://doi.org/10.1016/j.rse.2020.112238

Response: Yes, we agree with the comment that ECMWF soil moisture data is used in the SMOS L3 retrieval algorithm (Wigneron et al., 2021). We will consider the effect of ECMWF soil moisture on SMOS L3 soil moisture and add the following discussion in the revised manuscript.

"Besides, the ERA-Interim soil moisture from ECMWF is also used in the operational SMOS L3 SM retrieval algorithm. For a given pixel, the total TB is simulated as the sum of several fractions contribution ($F_{NO}$: nominal (bare soil, low vegetation), $F_{FO}$: forest, and others as urban, water, etc.), i.e. $TB_{total} = TB_{FNO} + TB_{FFO} + TB_{others}$ (Fernandez-Moran et al., 2017). SMOS L3 retrievals are computed only over a fraction of the pixel (the "dominant" fraction where SM retrieval is meaningful over certain surface types) (Fernandez-Moran et al., 2017; Wigneron et al., 2021). For the remaining fraction of the pixel, only the contributions of that to the total signal need to be estimated based on ECMWF ERA-Interim SM (0-7 cm) as auxiliary input but no SM retrievals are performed. Previous studies have evaluated ERA-Interim soil moisture over China and pointed that ERA-Interim soil moisture shows an overestimation (Yang et al., 2020; Ling et al., 2021). Therefore, the overestimated ECMWF ERA-Interim SM (0-7 cm) leads to the underestimation of forest $TB_{FFO}$ contribution, which further leads to the overestimation of $TB_{FNO}$ and to a dry bias in the retrieved SMOS L3 SM (as there is a negative correlation between brightness temperature and soil moisture (Rao et al., 2007))."

References

Wigneron, J.-P., Li, X., Frappart, F., Fan, L., Al-Yaari, A., De Lannoy, G., Liu, X., Wang, M., Le Masson, E., Moisy, C., 2021. SMOS-IC data record of soil moisture and L-VOD: historical development, applications and perspectives. Remote Sens. Environ. 254, 112238, https://doi.org/10.1016/j.rse.2020.112238.

Fernandez-Moran, R., Wigneron, J. P., De Lannoy, G., Lopez-Baeza, E., Parrens, M., Mialon, A., Mahmoodi, A., Al-Yaari, A., Bircher, S., Al Bitar, A., Richaume, P. and Kerr, Y.: A new calibration of the effective scattering albedo and soil roughness parameters in the SMOS SM

retrieval algorithm, Int. J. Appl. Earth Obs., 62, 27-38, https://doi.org/10.1016/j.jag.2017.05.013, 2017.

Yang, S., Li, R., Wu, T., Hu, G., Xiao, Y., Du, Y., Zhu, X., Ni, J., Ma, J., Zhang, Y., Shi, J. and Qiao, Y.: Evaluation of reanalysis soil temperature and soil moisture products in permafrost regions on the Qinghai-Tibetan Plateau, Geoderma, 377, 114583, https://doi.org/10.1016/j.geoderma.2020.114583, 2020.

Ling, X., Huang, Y., Guo, W., Wang, Y., Chen, C., Qiu, B., Ge, J., Qin, K., Xue, Y., and Peng, J.: Comprehensive evaluation of satellite-based and reanalysis soil moisture products using in situ observations over China, Hydrol. Earth Syst. Sci., 25, 4209–4229, https://doi.org/10.5194/hess-25-4209-2021, 2021.

K. S. RAO, GIRISH CHANDRA & P. V. NARASIMHA RAO (1987) The relationship between brightness temperature and soil moisture Selection of frequency range for microwave remote sensing, International Journal of Remote Sensing, 8:10, 1531-1545, http://dx.doi.org/10.1080/01431168708954795.

So should SMOS L4 be considered a remote sensing product or a modeled product? Please adapt the discussion according to my comment

Response: Root zone soil moisture can't be measured by remote sensing techniques directly due to the limited penetration depth. SMOS L4 is derived from a modified exponential filter method applied to SMOS L3 surface soil moisture. Though the exponential filter is a statistics-based method, we think SMOS L4 should be considered as a modeled product. Other comments

Please discuss results considering a similar study made in China by Fan et al., RSE 2022
DOI: 10.1016/j.rse.2022.113283
Response: We will add the following discussion in the revised manuscript.

"In the study by Fan et al., (2022), three root-zone soil moisture (RZSM) products (SMAP-L4 V6, ERA5-land V2, GLDAS-Noah V2.1) are evaluated over croplands of the Jiangsu province, which is close to the Huaibei plain. A fourth RZSM dataset is derived from the ESA CCI SSM using an exponential filter, as for SMOS L4. Overall, the four RZSM products underestimate the in situ observations with median bias values ranging from -0.04 for ERA5-Land to -0.08 $m^3$ $m^{-3}$ for SMAP L4. SMAP L4 also presents the lowest ubRMSE value. Regarding the correlation coefficient (R), ERA5-Land obtains the highest R, followed by SMAP L4, ESA CCI RZSM, and GLDAS_Noah. SMAP L4 has overall better performance than GLDAS_Noah in terms of all evaluation metrics except for the bias. These results contrast with those we obtained over the Huaibei plain. In this study, SMAP L4 and GLDAS_Noah both overestimate the in situ RZSM with a median bias of 0.033 $m^3$ $m^{-3}$. On the other hand, SMAP L4 has a larger R value and a smaller ubRMSE value (R=0.37, ubRMSE= 0.039 $m^3$ $m^{-3}$) than GLDAS_Noah (R = 0.35, ubRMSE = 0.043 $m^3$ $m^{-3}$), which is consistent with results drawn by Fan et al. (2022). In both studies, the in situ stations are mainly located in croplands. The changes in the sign of the bias could be attributed to differences in soil properties. In the Huaibei plain, the main soil type is lime concretion black

soil, the main characteristic of which is (1) soil stratification, (2) poor soil permeability and water retention capacity due to high clay content, (3) clay swell in wet period due to water absorption and shrinkage in dry period due to water loss. During drought, the cracks in the soil column increase and deepen, resulting in capillary water breakage and more water evaporation. During rainy periods or during irrigation, the soil absorbs water and swells, closing the cracks and preventing water infiltration of rainfall. Water is then mainly lost in the form of surface runoff. This could explain the small RZSM values ranging from 0.2 to 0.3 $m^3$ $m^{-3}$ observed in the Huaibei plain and the larger RZSM values ranging from 0.3 to 0.4 $m^3$ $m^{-3}$ observed in the Jiangsu. The larger precipitation amount in the Jiangsu province could be another reason."

References
Fan, L., Xing, Z., Lannoy, G. D., Frappart, F., Peng, J., Zeng, J., Li, X., Yang, K., Zhao, T., Shi, J., Ma, H., Wang, M., Liu, X., Yi, C., Ma, M., Tang, X., Wen, J., Chen, X., Wang, C., Wang, L., Wang, G. and Wigneron, J.-P.: Evaluation of satellite and reanalysis estimates of surface and root-zone soil moisture in croplands of Jiangsu Province, China, Remote Sensing of Environment, 282, https://doi.org/10.1016/j.rse.2022.113283, 2022.

Sentence: "Previous studies have illustrated that the VOD retrievals from SMOS may be noisy" is not objective. All VOD products can be considered as noisy, not only SMOS ones. It depends on location and product version (L2, L3 or SMOS-IC). Some SMAP, ASCAT, AMSR2 versions of VOD can be considered as much more noisy than SMOS VOD.
Usually **a sliding window smoothing technique** (T = 7 -30 days) should be used for all VOD products.

Li et al., 2020 https://doi.org/10.1016/j.rse.2020.112208

Response: Yes, we agree with the comment and we will delete the sentence in the revised manuscript.

---

## Author Comment (AC2)

Response to Reviewer #2's comments on the manuscript HESS-2023-33

The paper evaluates 7 global root-zone soil moisture products and one regional product over the Huai river basin in China for the period from April 2015 to March 2020. The products are evaluated against each other and against in situ measurements from 58 sensor profiles. The differences in the products and their performance against in situ measurements is related to the skill of the products' forcing inputs (precipitation, air temperature), model structure, and model parameters (soil texture).

The authors find that the GLDAS_CLSM product performs best overall and has the highest R and lowest ubRMSE values. Root-zone soil moisture is overestimated w.r.t. the in situ measurements in the 7 products based on land surface modeling, which is traced to an overestimation of the precipitation inputs, too much drizzle, underestimation of the air temperature inputs, and errors in the prescribed soil texture.

The paper addresses a topic of interest to HESS readers and presents some important results about the skill of several widely-used global root-zone soil moisture products and one regional product. Unfortunately, the paper falls short in several aspects, as described in the major comments below. The paper *may* be suitable for publication in HESS after major revisions. However, given the gaps and errors in the submitted manuscript, it is more realistic to REJECT the paper at this time and encourage resubmission later if the authors are willing and able to address the shortcomings. This gives the authors a better chance to produce a revised manuscript of sufficient quality without being constrained by the customary 2-month period for major revisions.

The authors thank the Reviewer #2 for her/his constructive and insightful comments that help us improve the quality of the manuscript. The original comments from Reviewer #2 are in black font, and our responses are in blue font.

RC2: 'Comment on hess-2023-33', Anonymous Referee #2, 31 Mar 2023

Major comments:
1) There are clear errors in the authors' description of the product characteristics that impact the interpretation of the results. Lines 26-27, also Lines 592-593: "[MERRA-2 and SMAP L4] are based on the same LSM and on the same surface meteorological forcing...". While both products use the Catchment model, and both use atmospheric forcing data generated with GEOS, the model versions and the forcing data are not the same. MERRA-2 uses an older version of the Catchment model than SMAP L4 (Reichle et al. 2017, 10.1175/JHM-D-17-0063.1; Reichle et al. 2019), including (but not limited to) the difference that soil parameters are based on FAO data in MERRA-2 and on HWSD in SMAP L4. Re. surface meteorological forcing, MERRA-2 uses an older GEOS version 5.12 at 0.5deg resolution, whereas the forcing data in SMAP L4 is from the GEOS weather analysis ("Forward Processing", or FP) system with evolving versions 5.17 to 5.25 and at 0.25-degree resolution. In MERRA-2, the CPCU precipitation is used in its native climatology whereas in SMAP L4 the CPCU precipitation is rescaled to an independent reference climatology.

Response: We agree with this comment, and we will make our best to revise the manuscript accordingly. Regarding the version of the CLSM, SMAP L4 uses a more recent version of CLSM comprising a new soil dataset (HWSD/ STATSGO2) and new pedotransfer functions accounting for the effect of soil organic matter on soil hydraulic and thermal properties (De Lannoy et al., 2014). MERRA-2 and SMAP L4 use different model background precipitation (GEOS-5 Forward Processing (FP) system for SMAP L4 and GEOS-5 Forward Processing system for Instrument Teams (FP-IT) system for MERRA-2) (Reichle et al., 2017; Reichle et al., 2019). In MERRA-2, the CPCU precipitation is used in its native climatology to correct the GEOS FP-IT model background precipitation whereas in SMAP L4 the CPCU precipitation is rescaled to the climatology of the Global Precipitation Climatology Project, version 2.2 (GPCPv2.2) pentad precipitation product before correcting the GEOS-5 FP system.

Lines 26-27 ("These products are based on the same LSM and on the same surface meteorological forcing generated from the National Aeronautics and Space Administration (NASA) GEOS-5") will be deleted.

Lines 592-593 ("The higher R value between SMAP L4 and MERRA-2 RZSM was attributed to the fact that SMAP L4 and MERRA-2 are both based on CLSM and on the same surface meteorological forcing generated from the NASA GEOS-5 in which precipitation was corrected with the gauge based CPCU precipitation product") will be deleted.

Reference:

De Lannoy, G. J. M., R. D. Koster, R. H. Reichle, S. P. P. Mahanama, and Q. Liu (2014), An updated treatment of soil texture and associated hydraulic properties in a global land modelling system, J. Adv. Model. Earth Syst., 6, 957–979, https://doi.org/10.1002/2014MS000330.

Reichle, R. H., Liu, Q., Koster, R. D., Draper, C. S., Mahanama, S. P. P. and Partyka, G. S.: Land Surface Precipitation in MERRA-2, J. Clim., 30, 1643-1664, https://doi.org/10.1175/jcli-d-16-0570.1, 2017.

Reichle, R. H., Liu, Q., Koster, R. D., Crow, W. T., De Lannoy, G. J. M., Kimball, J. S., Ardizzone, J. V., Bosch, D., Colliander, A., Cosh, M., Kolassa, J., Mahanama, S. P., Prueger, J., Starks, P. and Walker, J. P.: Version 4 of the SMAP Level-4 Soil Moisture Algorithm and Data Product, J. Adv. Model. Earth Syst., 11, 3106-3130, https://doi.org/10.1029/2019ms001729, 2019.

These differences are not discussed in the paper but are critical for the interpretation of the results, especially those re. errors in the products' soil parameters and the cause of the RZSM bias. For example, Figure 8 shows that SMAP L4 and MERRA-2 precipitation metrics vs in situ measurements are different, but the authors simply ignore this result and keep repeating that SMAP L4 and MERRA-2 have the same forcing data (e.g., Line 592). The fact that MERRA-2 also uses CMAP precipitation inputs is irrelevant (Line 493) because CMAP is used in MERRA-2 only over Africa and the ocean.

Response: We will add the following discussion in the revised manuscript.

Lines 489-496 (Section 5.3 "The very good correlation and low ubRMSE between MERRA-2 and SMAP L4 shown in Fig. 5 may be partly attributed to the fact that SMAP L4 and MERRA-2 share the same surface meteorological forcing generated from GEOS-5. Moreover, the SMAP L4 precipitation data generated by NASA GEOS-5 is corrected with the NOAA CPCU gauge-based analysis of global daily precipitation product. The MERRA-2 precipitation data are also corrected with CPCU but the Climate Prediction Center Merged Analysis of Precipitation (CMAP) product is used too. Since precipitation is the dominant driver of the land surface water cycle, this can explain the large R value between SMAP L4 and MERRA-2 RZSM products. In addition, both SMAP L4 and MERRA-2 use the CLSM.") will be replaced by:

"Regarding the intercomparison in section 4.2, the very good correlation and low ubRMSE between MERRA-2 and SMAP L4 shown in Fig. 5 may be partly attributed to the fact that both products are based on the CLSM and both use atmospheric forcing data generated with GEOS-5. However, SMAP L4 uses a more recent version of CLSM with a different representation of soil hydraulic and thermal properties. MERRA-2 and SMAP L4 use different model background precipitation (GEOS-5 Forward Processing (FP) system for SMAP L4 and GEOS-5 Forward Processing system for Instrument Teams (FP-IT) system for MERRA-2) (Reichle et al., 2017; Reichle et al., 2019). In MERRA-2, the CPCU precipitation is used in its native climatology to correct the GEOS FP-IT model background precipitation whereas in SMAP L4 the CPCU precipitation is rescaled to the climatology of the Global Precipitation Climatology Project, version 2.2 (GPCPv2.2) pentad precipitation product before correcting the GEOS-5 FP system. Figure 8 shows that SMAP L4 and MERRA-2 precipitation metrics vs in situ measurements are different. De Lannoy et al. (2014) showed that SMAP L4 has smaller mean bias of SSM and RZSM than MERRA-2 due to the increased sand content of HWSD and new pedotransfer functions provided by Wösten et al. (2001)."

Reference:

Reichle, R. H., Liu, Q., Koster, R. D., Draper, C. S., Mahanama, S. P. P. and Partyka, G. S.: Land Surface Precipitation in MERRA-2, J. Clim., 30, 1643-1664, https://doi.org/10.1175/jcli-d-16-0570.1, 2017.

Reichle, R. H., Liu, Q., Koster, R. D., Crow, W. T., De Lannoy, G. J. M., Kimball, J. S., Ardizzone, J. V., Bosch, D., Colliander, A., Cosh, M., Kolassa, J., Mahanama, S. P., Prueger, J., Starks, P. and Walker, J. P.: Version 4 of the SMAP Level-4 Soil Moisture Algorithm and Data Product, J. Adv. Model. Earth Syst., 11, 3106-3130, https://doi.org/10.1029/2019ms001729, 2019.

De Lannoy, G. J. M., R. D. Koster, R. H. Reichle, S. P. P. Mahanama, and Q. Liu (2014), An updated treatment of soil texture and associated hydraulic properties in a global land modelling system, J. Adv. Model. Earth Syst., 6, 957–979, https://doi.org/10.1002/2014MS000330.

Wösten J., Y. A. Pachepsky, and W. Rawls (2001), Pedotransfer functions: Bridging the gap between available basic soil data and missing soil hydraulic characteristics, J. Hydrol., 251, 123–150, https://doi.org/10.1016/S0022-1694(01)00464-4.

I outlined the above differences between MERRA-2 and SMAP L4 in detail because I am

most familiar with these products. I did not exhaustively review the accuracy of the other products' characteristics. Descriptions of the other products may or may not contain additional errors.

Response: We have checked the other products and corrected the additional errors in the revised manuscript.

In the last column in Table 1, for the SMAP L4 data access, "SMAP L4 Global 3-hourly 9 km EASE-Grid Surface and Root Zone Soil Moisture Analysis Update, Version 5 | National Snow and Ice Data Center (nsidc.org)" will be replaced by "SMAP L4 Global 3-hourly 9 km EASE-Grid Surface and Root Zone Soil Moisture Geophysical Data, Version 5 | National Snow and Ice Data Center (nsidc.org)".

2) Besides the errors in the product descriptions outlined above, there is insufficient information in the description of the products and analysis methods:
a. Section 2.3 is very uneven across the subsections (individual product descriptions) in terms of the level of detail provided. E.g., in some cases the horizontal resolution of the product is not provided.

Response: We will reword the description of each product in the section 2.3. The horizontal resolution of the products is provided in Table 1.

b. For MERRA-2 the native 0.5-deg data is interpolated to 0.25 degree, but there is no comparable statement for SMAP L4. Are the SMAP L4 data used in their native (9-km) resolution? Aggregated to 0.25 degree

Response: Yes, MERRA-2 has the native resolution of 0.625-degree longitude by 0.5-degree latitude, which was interpolated to 0.25-degree (GLDAS-2_0.25 grid) by GES DISC (https:/disc.gsfc.nasa.gov/datasets?project=MERRA-2). For the SMAP L4, we used the native resolution (9-km) and did not aggregate it to 0.25-degree. We did not change the resolution of any products ourselves.

c. ERA5 includes a soil moisture and screen-level temperature and humidity analysis. This analysis clearly impacts the ERA5 soil moisture estimates but is not mentioned even once in the paper.

Response: We will add the following discussion in the revised manuscript.

"ERA5 includes a soil moisture and screen-level (2 m) air temperature and air humidity analysis. Studies indicated that the assimilation of screen-level variables improves root zone soil moisture estimates against in situ observations and provides a more realistic lower boundary conditions for numerical prediction model (Douville et al., 1999; Seffert et al., 2003; De Rosnay et al., 2013)."

Reference:

Seuffert, G., H. Wilker, P. Viterbo, J.-F. Mahfouf, M. Drusch, J.-C. Calvet., 2003: Soil moisture analysis combining screen-level parameters and microwave brightness temperature: A test with field data. Geophys. Res. Lett., **30**, 1498, https://doi.org/10.1029/2003GL017128.

Douville, H., P. Viterbo, J.-F. Mahfouf, and A. Beljaars, 2000: Evaluation of the optimum interpolation and nudging technique for soil moisture analysis using FIFE data. Mon. Wea. Rev., 128, 1733–1756, https://doi.org/10.1175/15200493(2000)128<1733:EOTOIA>2.0.CO;2.

De Rosnay, P., Drusch, M., Vasiljevic, D., Balsamo, G., Albergel, C., and Isaksen, L., 2013: A simplified Extended Kalman Filter for the global operational soil moisture analysis at

ECMWF, Q. J. Roy. Meteorol. Soc., 139, 1199–1213, https://doi.org/10.1002/qj.2023.

d. Much of the text in the subsections of section 2.3 appears copied from the blurb of the descriptions on the products' websites. In some cases, the text includes to the motivation of products' development from the preceding version. E.g., MERRA-2 is described in reference to the long-obsolete original MERRA data (Lines 159-163), and NCEP CFS v2.2 is described in reference to NCEP R1 and R2 (~Line 176). The product descriptions in this paper should focus on each product's characteristics and how they differ from the other products examined here, not on how the products differ from older versions that are not examined here.

Response: We will reword the section 2.3 in the revised manuscript. The section 2.3 will be replaced by the following content:

**2.3.1 ERA5**

ERA5 is the fifth generation global atmospheric reanalysis produced by ECMWF. It covers the period from January 1940 to present, and provides hourly, 0.25-degree atmosphere, land surface and 0.5-degree ocean waves estimates (Hersbach et al., 2023). ERA5 is developed using 4-Dimensional Variational (4D-Var) data assimilation with an underlying 10-member ensemble and model forecasts from the CY41R2 version of the ECMWF Integrated Forecast System (IFS), with 137 hybrid sigma/pressure model levels in the vertical and the top level at 0.01 hPa (Hersbach et al., 2020; Xu et al., 2021). The 4D-Var data assimilation uses 12 hour windows from 0900 UTC to 2100 UTC and from 2100 UTC to 0900 UTC (the following day). The HTESSEL scheme is used as the land surface component of ERA5 to model land surface variables. The data assimilation is based on the Simplified Extended Kalman Filter (SEKF) for RZSM, 1-dimensional optimal interpolation (OI) for soil and snow temperature, and 2-dimensional OI for snow and screen-level parameters (2 m temperature and relative humidity) (Hersbach et al., 2020). The screen-level parameters analysis is carried out first, then its increments are used as input for the soil moisture analysis.

**2.3.2 MERRA-2**

MERRA-2 is the latest version of a global atmospheric reanalysis product produced by the NASA Global Modeling and Assimilation Office (GMAO). It uses the Goddard Earth

Observing System Model (GEOS-5.12.4) atmospheric data assimilation system composed of (1) the GEOS atmospheric model and (2) the Gridpoint Statistical Interpolation assimilation system. It covers the period from January 1980 to present with a latency of ~3 weeks after the end of a month and provides global, hourly, 0.25-degree estimates (GMAO, 2015; Reichle et al., 2017c). CLSM is used as the land surface component of MERRA-2 to analyze the land surface states and fluxes. The precipitation forcing is the weighted average of model background precipitation generated by GEOS-5 FP-IT (Forward Processing system for Instrument Teams) after 31 December 2014 and precipitation generated by AGCM, and the weights are dependent on latitude. The National Oceanic and Atmospheric Administration (NOAA) Climate Prediction Center (CPC) Unified Gauge-Based Analysis of Global Daily Precipitation (CPCU) product is used to correct model background precipitation. The CPC Merged Analysis of Precipitation (CMAP) product is rescaled to match the climatology of Global Precipitation Climatology Project product, version 2.1 (GPCPv2.1) and used fully in Africa, which allows the observed precipitation to impact, via evapotranspiration, the near-surface air temperature and humidity, thereby yielding a more self-consistent near-surface meteorological dataset (Reichle et al., 2017c).

**2.3.3 NCEP CFSv2**

NCEP CFSv2 is a global, high resolution, coupled atmosphere-ocean-land surface-sea ice system designed to provide the best estimate of the state of these coupled domains, it covers the period from January 2011 to present and provides 6-hourly, 0.2-degree estimates (Saha, 2011). The Noah land surface model is used in the semi-coupled Climate Forecast System Reanalysis (CFSR) Global Land Data Assimilation System (GLDAS) for providing the land surface analysis and evolving land surface states (Saha et al., 2010; Saha et al., 2014).

**2.3.4 GLDAS_NOAH**

GLDAS_NOAH Version 2.1 provides global, 3-hourly, 0.25-degree resolution of estimates covering the period from 1 January 2000 to present (Rodell et al., 2004; Beaudoing et al., 2020). The offline (not coupled to the atmosphere) Noah LSM is forced with combination of model- (NOAA/Global Data Assimilation System (GDAS) atmospheric analysis fields) and observation-based precipitation (the disaggregated Global Precipitation Climatology Project (GPCP) V1.3 Daily Analysis precipitation fields) and radiation data (the Air Force Weather Agency's AGRicultural METeorological modeling system (AGRMET) radiation fields) to provide optimal fields of land surface analysis (Rui et al., 2021).

**2.3.5 GLDAS_CLSM**

GLDAS_CLSM Version 2.2 is based on the CLSM forced with the meteorological analysis fields from the operational ECMWF IFS and provides global, daily, 0.25-degree resolution estimates covering the period from 1 February 2003 to present. (Li et al., 2019; Li et al., 2020; Rui et al., 2021). GLDAS-2.2 assimilates the total terrestrial water (TWS) anomaly observations from Gravity Recovery and Climate Experiment (GRACE). The temporal changes of TWS are influenced by changes in soil moisture, snow and ice, surface water and biomass, and ground water storage.

**2.3.6 CLDAS**

The CLDAS-2.0 product is developed and released by CMA based on a multi-LSMs operational system consisting of CLM, CoLM, and Noah-MP, with a spatial coverage of 0-60° N and 70-150° E and temporal coverage from January 2008 to present (CMA, 2015). The production of CLDAS-V2.0 includes the following three processes. Firstly, nearly 40000 automatic meteorological stations measurements, ECMWF and NCEP numerical analysis/forecast product, satellite-derived precipitation (FY2) and Digital Elevation Model (DEM) are used to produce 0.0625°, hourly estimates of meteorological forcing data by operating the Space-Time Multi-Scale Analysis System (STMAS) (Shi et al., 2014; Wang et al., 2021a). Meantime, the meteorological forcing is validated using national automatic station observations (more than 2400 stations). Secondly, the meteorological forcing is used to drive the multi-LSMs system for obtaining a multilayer soil moisture estimates ensemble. Finally, ensemble-average is applied to each soil layer to generate a soil moisture ensemble analysis product.

**2.3.7 SMAP L4**

The SMAP Level-4 soil moisture (L4-SM) is produced by assimilating SMAP radiometer level-1C brightness temperature observations into CLSM and provides global, 3-hourly, 9-km resolution estimates of SSM (0-5 cm) and RZSM (0-100 cm) from March 2015 to present (Reichle et al., 2020; Reichle et al., 2021). The Goddard Earth Observation System, version 5, LDAS (GEOS-5 LDAS) uses a spatially distributed ensemble Kalman filter (EnKF) to assimilate the observations into CLSM (Rienecker et al., 2008). The EnKF has a 3-hourly

update time step and is used to interpolate and extrapolate the brightness temperature and model estimates in time and space (Reichle et al., 2017a). The GEOS-5 CLSM is driven by surface meteorological data (precipitation, radiation, etc.) from the GEOS-5 Forward Processing (FP) system where large amounts of observations are assimilated into a global atmospheric model. The CPCU, 0.5-degree, daily precipitation observations are used for correcting the GEOS-5 FP model background precipitation. Prior to the GEOS-5 FP precipitation correction, the CPCU precipitation data and hourly background precipitation are both scaled to the climatology of the GPCPv2.2 pentad precipitation product.

**2.3.8 SMOS L4**

The SMOS L4 soil moisture product is disseminated by SMOS CATDS and provides global, daily estimates of RZSM (0–100 cm) over a 25-km EASE-2 grid from January 2010 to present (Al Bitar and Mahmoodi, 2020; CATDS, 2021). The SMOS L4 RZSM is derived from the SMOS L3 3-day SSM product using a modified exponential filter linking the characteristic time length T (the transfer time for water from surface layer to root zone layer) to the soil properties (Pablos et al., 2018). The product is based on SMOS descending orbit (18:00) observations and on other ancillary datasets, such as MODIS observations, climate data from the NCEP, and an upgraded FAO/UNESCO soil properties map. The soil column is divided into three layers (layer 1: 0-5 cm, layer 2: 5-40 cm, layer 3: 40-100 cm) in a water bucket model. The scaled 0-5 cm soil moisture is modified using a logarithmic function and filtered to obtain the layer 2 soil moisture. Then the scaled layer 2 soil moisture is filtered using another value of T to obtain the layer3 soil moisture. Finally, the RZSM (0-100 cm) is computed as a depth-weighted average of the three layers' soil moisture (Al Bitar et al., 2021).

e. The information on the specific air and soil temperature variables provided in the manuscript is not sufficient. What are the air temperatures that are used in the comparison? Are they at the 2-m screen level (T2m) or at the atmospheric model's lowest level? The SMAP L4 product only provides the latter, which is used to drive the Catchment model.

Response: In this study, in situ air temperature observations at 2-m are used for comparison with the air temperature values from the products. It must be noticed that SMAP L4 provides the air temperature at center height of the lowest atmospheric model layer, not at 2 m.

f. As for the simulated soil temperature, what is the layer depth for each product? And what exactly is the in situ soil temperature shown in Fig 9? The strong short-term variability of the in situ soil temperature (after averaging across 58 stations!) is highly suspect. Could this be the surface (or skin) temperature? The comparison here is almost certainly one of apples vs. oranges.

Response: We think Reviewer 2 misinterpreted Fig. 9. In this Figure, a comparison between simulated air temperature products and gridded air temperature dataset reference is shown.

g. What is the spatial resolution at which the temperature is validated?

Response: The air temperature has same spatial resolution as soil moisture (refer to Table 1).

h. The in situ soil moisture measurements are taken at 8am local time (Line 109), but what about the in situ soil temperature measurements? If the in situ soil temperature is taken at 8 am local time, are the soil temperatures from the products (Fig 9) also sampled at 8am local time? Are daily average soil temperatures from the products used?

Response: The in situ surface soil temperature measurements are measured at local time (08:00 am, 12:00 am and 20:00 PM) by thermometer. Daily average soil temperatures from the products are compared with daily average value of in situ surface soil temperature measurements in Figure S2. For SMOS L4, the surface skin temperatures observed at 20:00 PM are compared with soil temperature of SMOS L3 (descending orbit, 18:00 PM). Fig. 9 is the comparison between simulated air temperature products (daily average value) and gridded air temperature dataset reference (daily average value). This will be made clear in the caption of Fig. 9.

i. It is not clear at what time step the second-order skill metrics (R, ubRMSE) are computed. Are the metrics computed at daily time steps after aggregating all sub-daily products to daily time steps?

Response: Yes, all the metrics (including R and ubRMSE) are calculated on a daily basis (after aggregating all sub-daily products to daily time steps). This will be made clear in the text.

j. The in situ soil moisture data are QC'd (Lines 111-112) but there is no information on how many data points are actually used at each station. The footnote of Table 2 suggests that 1827 (daily?) values are used at each of the 58 stations, but then no data would have been excluded by the QC process. This is contradictory.

Response: We will revise the sentence and add a new Table S1 (see below) in the supplement. In this study, daily observations are considered for a time period of 1827 days from 1 April 2015 to 31 March 2020. The number of missing data across the 58 stations varies from 15 to 75. We gap-filled the observations at each station by taking the observations from the day before. so that 1827 observations are used.

The footnote of Table 2 will be replaced by "n is the number of gap-filled daily observations (1827) used at each of the 58 in situ stations (see Table S1)"

Table S1 Distribution of Huai River Basin in situ stations and observed points

| Station ID | Station Name | Longitude (E) | Latitude (N) | Elevation (m) | Time series length of the study period (day) | Number of soil moisture observations (day) | Number of missing soil moisture observations (day) |
|---|---|---|---|---|---|---|---|
| 50402241 | Taolaoba | 117.164 | 32.184 | 48 | 1827 | 1809 | 18 |
| 50403609 | Chahua | 116.022 | 33.033 | 39 | 1827 | 1804 | 23 |
| 50403809 | Hanting | 116.319 | 33.021 | 28 | 1827 | 1807 | 20 |
| 50420400 | Songji | 115.271 | 32.815 | 39 | 1827 | 1802 | 25 |
| 50421000 | Funan | 115.571 | 32.637 | 33 | 1827 | 1776 | 51 |
| 50421800 | Santa | 115.697 | 32.808 | 33 | 1827 | 1776 | 51 |
| 50423201 | Yaoli | 116.172 | 31.823 | 58 | 1827 | 1752 | 75 |
| 50424701 | Guanting | 116.851 | 31.797 | 51 | 1827 | 1808 | 19 |
| 50426001 | Zhuangmu | 117.112 | 32.363 | 27 | 1827 | 1812 | 15 |
| 50426072 | Guiji | 116.623 | 32.778 | 23 | 1827 | 1807 | 20 |
| 50426801 | Xiaji | 116.540 | 32.654 | 25 | 1827 | 1809 | 18 |
| 50429700 | Shuangfu | 115.569 | 33.342 | 37 | 1827 | 1775 | 52 |
| 50430100 | Fentai | 115.727 | 33.455 | 35 | 1827 | 1776 | 51 |
| 50430117 | Santang | 115.829 | 33.314 | 32 | 1827 | 1801 | 26 |
| 50430709 | Lixin | 116.209 | 33.143 | 28 | 1827 | 1780 | 47 |
| 50601600 | Jieshou | 115.359 | 33.265 | 42 | 1827 | 1776 | 51 |
| 50609001 | Yangqiao | 115.392 | 33.017 | 28 | 1827 | 1775 | 52 |
| 50634550 | Guangwu | 115.334 | 33.374 | 42 | 1827 | 1802 | 25 |
| 50636750 | Huangling | 115.134 | 33.041 | 37 | 1827 | 1776 | 51 |
| 50637371 | Quanyang | 115.437 | 33.112 | 35 | 1827 | 1801 | 26 |
| 50637413 | Kanheliu | 115.852 | 33.099 | 33 | 1827 | 1775 | 52 |
| 50637427 | Kouziji | 116.087 | 32.844 | 26 | 1827 | 1776 | 51 |
| 50637450 | Sanshilipu | 116.106 | 32.697 | 27 | 1827 | 1775 | 52 |
| 50637459 | Xiaqiao | 116.384 | 32.643 | 26 | 1827 | 1774 | 53 |
| 50700401 | Hengpaitou | 116.364 | 31.590 | 72 | 1827 | 1769 | 58 |
| 50701303 | Xianghongdianx | 116.177 | 31.580 | 116 | 1827 | 1779 | 48 |
| 50725311 | Wangchenggan | 116.526 | 31.740 | 76 | 1827 | 1784 | 43 |
| 50830409 | Lumiao | 115.795 | 33.998 | 39 | 1827 | 1776 | 51 |
| 50830419 | Dasi | 115.873 | 33.802 | 42 | 1827 | 1778 | 49 |
| 50830439 | Youhe | 115.789 | 33.631 | 38 | 1827 | 1803 | 24 |

| 50830449 | Huagou | 116.063 | 33.510 | 33 | 1827 | 1811 | 16 |
|---|---|---|---|---|---|---|---|
| 50830480 | Dahu | 116.351 | 33.515 | 31 | 1827 | 1809 | 18 |
| 50830489 | Chenqiao | 116.561 | 33.094 | 25 | 1827 | 1777 | 50 |
| 50830601 | Heliu | 116.967 | 33.033 | 25 | 1827 | 1809 | 18 |
| 50900601 | Linhuanzha | 116.567 | 33.667 | 29 | 1827 | 1809 | 18 |
| 50901501 | Guzhenzha | 117.333 | 33.300 | 18 | 1827 | 1811 | 16 |
| 50903176 | Wudaogou | 117.341 | 33.156 | 21 | 1827 | 1808 | 19 |
| 50903421 | Hexiangzha | 117.183 | 33.000 | 18 | 1827 | 1806 | 21 |
| 50903600 | Tancheng | 116.557 | 33.441 | 29 | 1827 | 1804 | 23 |
| 50903541 | Xibakou | 117.867 | 33.150 | 11 | 1827 | 1809 | 18 |
| 50907801 | Xulouzha | 116.750 | 33.917 | 30 | 1827 | 1812 | 15 |
| 50908001 | Suxianzha | 117.083 | 33.667 | 28 | 1827 | 1811 | 16 |
| 50909701 | Gukouzha | 116.450 | 34.267 | 39 | 1827 | 1810 | 17 |
| 50912201 | Kuaitanggou | 117.550 | 33.750 | 20 | 1827 | 1810 | 17 |
| 50913201 | Yanglou | 116.783 | 34.317 | 39 | 1827 | 1809 | 18 |
| 50913901 | Langanji | 117.233 | 33.934 | 25 | 1827 | 1811 | 16 |
| 50922032 | Dulou | 116.850 | 34.200 | 37 | 1827 | 1811 | 16 |
| 50922072 | Xiangyang | 117.583 | 33.467 | 24 | 1827 | 1809 | 18 |
| 50922172 | Shuangdui | 116.900 | 33.417 | 25 | 1827 | 1811 | 16 |
| 50922232 | Shuoli | 116.900 | 34.033 | 32 | 1827 | 1812 | 15 |
| 50922332 | Huangmiao | 117.652 | 33.079 | 19 | 1827 | 1810 | 17 |
| 50924801 | Baoji | 117.113 | 33.158 | 22 | 1827 | 1807 | 20 |
| 50925801 | Dinghouying | 117.338 | 33.457 | 24 | 1827 | 1811 | 16 |
| 50931578 | Xuanmiao | 116.267 | 34.517 | 54 | 1827 | 1810 | 17 |
| 50932801 | Longhai | 116.350 | 34.400 | 45 | 1827 | 1811 | 16 |
| 50933001 | Zhangzhuangzh | 116.600 | 34.117 | 37 | 1827 | 1811 | 16 |
| 50935201 | Sixian | 117.917 | 33.434 | 16 | 1827 | 1811 | 16 |
| 50938101 | Dazhuang | 117.867 | 33.667 | 20 | 1827 | 1812 | 15 |

3) Many references are plainly incorrect or inappropriate, or are missing altogether:

a. Many of the references used for the products are not the first-hand references. Rather, the references used are simply about applications of the data product. This is not acceptable. The authors must use first-hand references for the data products examined. In other cases, inappropriate references are cited, or references are simply wrong. Here are some of the problematic references (not meant to be a complete list):

Response: We will check all the references.

i. Bi et al. 2016 is used in several places as a reference for GLDAS product characteristics but the paper is not a first-hand description of the GLDAS products.

Response: We will replace the reference with the following one.

Rodell, M., Houser, P. R., Jambor, U., Gottschalck, J., Mitchell, K., Meng, C. J., Arsenault, K., Cosgrove, B., Radakovich, J., Bosilovich, M., Entin, J. K., Walker, J. P., Lohmann, D. and Toll, D.: The Global Land Data Assimilation System, B. Am. Meteorol. Soc., 85, 381-394, https://doi.org/10.1175/bams-85-3-381, 2004.

ii. Line 51: The main ERA5 reference is Hersbach et al. QJRMS 2020.

Response: we will replace the reference with the following one.

Hersbach, H., Bell, B., Berrisford, P., Hirahara, S., Horányi, A., Muñoz-Sabater, J., Nicolas, J., Peubey, C., Radu, R., Schepers, D., Simmons, A., Soci, C., Abdalla, S., Abellan, X., Balsamo, G., Bechtold, P., Biavati, G., Bidlot, J., Bonavita, M., Chiara, G., Dahlgren, P., Dee, D., Diamantakis, M., Dragani, R., Flemming, J., Forbes, R., Fuentes, M., Geer, A., Haimberger, L., Healy, S., Hogan, R. J., Hólm, E., Janisková, M., Keeley, S., Laloyaux, P., Lopez, P., Lupu, C., Radnoti, G., Rosnay, P., Rozum, I., Vamborg, F., Villaume, S. and Thépaut, J. N.: The ERA5 global reanalysis, Quarterly Journal of the Royal Meteorological Society, 146, 1999-2049, https://doi.org/10.1002/qj.3803, 2020.

iii. Line 46: Rienecker et al 2008 is not an appropriate reference for SMAP L4.

Response: we will replace the reference with the following one.

Reichle, R., Crow, W., Koster, R., Kimball, J. and Lannoy, G. D.: Algorithm Theoretical Basis Document (ATBD) SMAP Level 4 Surface and Root Zone Soil Moisture (L4_SM) Data Product, Soil Moisture Active Passive (SMAP) Project, Available at https://smap.jpl.nasa.gov/files/smap2/L4_SM_InitRel_v1.pdf, 2012.

iv. Line 56: Reichle et al. 2017 is not an appropriate reference for NCEP CFSv2.

Response: we will delete the reference.

v. Line 80: Koster et al. 2016 is really McCarty et al. 2016.

Response: we will modify the reference as the following one.

McCarty, W., Coy, L., Gelaro, R., Huang, A., Merkova, D., Smith, E. B., Sienkiewicz, M. and Wargan, K.: MERRA-2 Input Observations: Summary and Assessment NASA Tech. Rep. Series on Global Modeling and Data Assimilation. 46, 1-64, https://gmao.gsfc.nasa.gov/pubs/docs/McCarty885.pdf, 2016.

vi. Lines 83-84: Need a reference for this statement about "layer stratification" (and also edit the statement for clarity); also in Lines 479-480.

Response: We will add the reference with the following one.

Li, D., Zhang, g. and Gong, z.: On Taxonomy of Shajiang Black Soils in China, Soils, 43, 623-629,2011. Lorenz, R., Jaeger, E. B. and Seneviratne, S. I.: Persistence of heat waves and its link to soil moisture memory, Geophys. Res. Lett., 37, L09703, https://doi.org/10.1029/2010gl042764, 2010.

Zha, L., Wu, K., Li, L., Chen, J. and Ju, B.: The Cultivation Obstacle Factors of Lime Concretion Black Soil Genuses in Henan (in Chinese with English abstract) Chinese Journal of Soil Science, 46, 280-286, https://doi.org/10.19336/j.cnki.trtb.2015.02.004, 2015.

Gu, F., Chen, X., Wei, C., Zhou, M. and Li, B.: Distribution of calcareous concretion in soil profile and their effects on soil water retention in calcic vertisol (in Chinese with English abstract) Transactions of the Chinese Society of Agricultural Engineering, 37, 73-80, https://doi.org/10.11975/j.issn.1002-6819.2021.06.010, 2021.

b. There is no reference for the HWSD soil data!

Response: we will add the reference with the following one.

FAO, IIASA, ISRIC, ISSCAS and JRC: Harmonized World Soil Database (version 1.2), Feb 2012, Available at http://webarchive.iiasa.ac.at/Research/LUC/External-World-soil-database/HWSD_Documentation.pdf, 2012.

c. References by the same lead author written in the same year are not distinguished

properly. E.g., there are two papers by Wang et al. 2016 and three papers by Reichle et al. 2017 in the list of references. In the text, they are universally referred to as Wang et al. 2016 or Reichle et al. 2017, leaving the reader guessing which paper the authors refer to in each instance.

Response: we have added English letters (a, b, c..) to distinguish all the references by the same lead author written in the same year. E.g., Want et al. 2016a and Want et al. 2016b; Reichle et al. 2017a, Reichle et al. 2017b and Reichle et al. 2017c.

Wang, X., Lü, H., Crow, W. T., Zhu, Y., Wang, Q., Su, J., Zheng, J. and Gou, Q.: Assessment of SMOS and SMAP soil moisture products against new estimates combining physical model, a statistical model, and in-situ observations: A case study over the Huai River Basin, China, J. Hydro., 598, 126468, https://doi.org/10.1016/j.jhydrol.2021.126468, 2021a.

Wang, Z., Che, T., Zhao, T., Dai, L., Li, X. and Wigneron, J.-P.: Evaluation of SMAP, SMOS, and AMSR2 Soil Moisture Products Based on Distributed Ground Observation Network in Cold and Arid Regions of China, IEEE J-STARS, 14, 8955-8970, https://doi.org/10.1109/jstars.2021.3108432, 2021b.

Reichle, R., G. De Lannoy, R. D. Koster, W. T. Crow, J. S. Kimball and Liu, Q.: SMAP L4 Global 3-hourly 9 km EASE-Grid Surface and Root Zone Soil Moisture Geophysical Data, Version 5 [Data Set], Boulder, Colorado USA. NASA National Snow and Ice Data Center Distributed Active Archive Center, https://doi.org/10.5067/9LNYIYOBNBR5. Date Accessed 06-04-2021, 2020.

Reichle, R. H., De Lannoy, M., G. J. and Liu, Q.: Assessment of the SMAP Level-4 Surface and Root-Zone Soil Moisture Product Using In Situ Measurements, J. Hydrometeorol., 18, 2621-2645, https://doi.org/10.1175/jhm-d-17-0063.1, 2017a.

Reichle, R. H., De Lannoy, G. J. M., Liu, Q., Koster, R. D., Kimball, J. S., Crow, W. T., Ardizzone, J. V., Chakraborty, P., Collins, D. W., Conaty, A. L., Girotto, M., Jones, L. A., Kolassa, J., Lievens, H., Lucchesi, R. A. and Smith, E. B.: Global Assessment of the SMAP Level-4 Surface and Root-Zone Soil Moisture Product

Using Assimilation Diagnostics, J. Hydrometeorol., 18, 3217-3237, https://doi.org/10.1175/JHM-D-17-0130.1, 2017b.

Reichle, R. H., Liu, Q., Koster, R. D., Draper, C. S., Mahanama, S. P. P. and Partyka, G. S.: Land Surface Precipitation in MERRA-2, J. Clim., 30, 1643-1664, https://doi.org/10.1175/jcli-d-16-0570.1, 2017c.

d. Data products such as SMAP L4 that have a digital object identifier must be cited with proper references that include the DOI.

Response: We have added the following references of all products in the revised manuscript.

GLDAS_NOAH:

Beaudoing, H. and M. Rodell, NASA/GSFC/HSL (2020), GLDAS Noah Land Surface Model L4 3 hourly 0.25 x 0.25 degree V2.1, Greenbelt, Maryland, USA, Goddard Earth Sciences Data and Information Services Center (GES DISC), Accessed: [*21 September 2021*], 10.5067/E7TYRXPJKWOQ.

Rodell, M., P.R. Houser, U. Jambor, J. Gottschalck, K. Mitchell, C. Meng, K. Arsenault, B. Cosgrove, J. Radakovich, M. Bosilovich, J.K. Entin, J.P. Walker, D. Lohmann, and D. Toll, 2004: The Global Land Data Assimilation System, Bull. Amer. Meteor. Soc., 85, 381-394, doi:10.1175/BAMS-85-3-381.

GLDAS_CLSM:

Li, B., H. Beaudoing, and M. Rodell, NASA/GSFC/HSL (2020), GLDAS Catchment Land Surface Model L4 daily 0.25 x 0.25 degree GRACE-DA1 V2.2, Greenbelt, Maryland, USA, Goddard Earth Sciences Data and Information Services Center (GES DISC), Accessed: [*22 September 2021*], 10.5067/TXBMLX370XX8.

Li, B., M. Rodell, S. Kumar, H. Beaudoing, A. Getirana, B. F. Zaitchik, et al. (2019) Global GRACE data assimilation for groundwater and drought monitoring: Advances and challenges. Water Resources Research, 55, 7564-7586. doi:10.1029/2018wr024618.

ERA5:

Hersbach, H., Bell, B., Berrisford, P., Biavati, G., Horányi, A., Muñoz Sabater, J., Nicolas, J., Peubey, C., Radu, R., Rozum, I., Schepers, D., Simmons, A., Soci, C., Dee, D., Thépaut, J-N.: ERA5 hourly data on single levels from 1940 to present, Copernicus Climate Change Service (C3S) Climate Data Store (CDS), https://doi.org/10.24381/cds.adbb2d47, (Accessed: 22 September 2021), 2023.

MERRA-2:

Global Modeling and Assimilation Office (GMAO) (2015), MERRA-2 tavg1_2d_lnd_Nx: 2d,1-Hourly,Time-Averaged,Single-Level,Assimilation,Land Surface Diagnostics V5.12.4, Greenbelt, MD, USA, Goddard Earth Sciences Data and Information Services Center (GES DISC), Accessed: 26 September 2021, 10.5067/RKPHT8KC1Y1T.

NCEP CFSv2:

Saha, S., et al. 2011, updated monthly. *NCEP Climate Forecast System Version 2 (CFSv2) Selected Hourly Time-Series Products*. Research Data Archive at the National Center for Atmospheric Research, Computational and Information Systems Laboratory. https://doi.org/10.5065/D6N877VB. Accessed 28 October 2021.

SMAP L4:

Reichle, R., G. De Lannoy, R. D. Koster, W. T. Crow, J. S. Kimball, and Q. Liu. (2020). SMAP L4 Global 3-hourly 9 km EASE-Grid Surface and Root Zone Soil Moisture

Geophysical Data, Version 5 [Data Set]. Boulder, Colorado USA. NASA National Snow and Ice Data Center Distributed Active Archive Center. https://doi.org/10.5067/9LNYIYOBNBR5. Date Accessed 06-04-2021.

SMOS

CATDS (2021), CATDS-PDC L4SM RZSM – 1 day global map of root zone soil moisture values from SMOS satellite. CATDS (CNES, IFREMER, CESBIO). http://dx.doi.org/10.12770/316e77af-cb72-4312-96a3-3011cc5068d4. Date Accessed 17-09-2021

Al Bitar Ahmad, & Mahmoodi Ali. (2020, November 30). Algorithm Theoretical Basis Document (ATBD) for the SMOS Level 4 Root Zone Soil Moisture (Version v30_01). Zenodo. http://doi.org/10.5281/zenodo.4298572.

CLDAS:

CMA (2020), The near-real-time product dataset of the China Meteorological Administration Land Data Assimilation System (CLDAS-V2.0). Available at http://data.cma.cn/en/?r=search/uSearch&keywords=cldas. Date Accessed 16-11-2021.

4) The version of some of the products examined here are unclear. E.g., what SMAP L4 version is used in the paper? This has important implications on the interpretation of the results. The precipitation forcing data in SMAP L4 versions 5 and 6 differs considerably, which is well documented in the products' validation reports (disseminated by NSDIC along with the data).

Response: In this study, SMAP L4 Global 3-hourly 9 km EASE-Grid Surface and Root Zone Soil Moisture Geophysical Data (SPL4SMGP), version 5 is used in the paper. For SMAP L4 Version5, the precipitation forcing is corrected with CPCU precipitation, for SMAP L4 Version6, the precipitation forcing is corrected with Version 06B of IMERG-Late and IMERG-Final precipitation products. We will add the version information of the products in revised Table 1.

SMOS L4 RZSM Version301

GLDAS_NOAH Version2.1

GLDAS_CLSM Version2.2

CLDAS Version2.0

MERRA-2 Version2.0 (GEOS V5.12.4)

NCEP CFSv2 Version2.0

ERA5 hourly data on single levels from 1979 to present

5) Like SMAP L4 and CLSM, the GLDAS_CLSM product also uses the Catchment model and precipitation forcing based on observations, but there are clear differences in the skill metrics between the three products. One distinguishing feature of GLDAS_CLSM is the assimilation of GRACE TWS observations, which inherently contain information about root-zone soil moisture.

There is no discussion in the text on what the impact of the GRACE data assimilation may be on the skill of GLDAS_CLSM vs. the other two Catchment-based products. (By the way, the Catchment model versions in GLDAS_CLSM and MERRA-2 are different but closer to each other than to the version used in SMAP L4.)

Response: We will discuss the effect of GRACE TWS data assimilation on accuracy of RZSM over HRB add the following discussion in the revised manuscript.

"Regarding the in situ validation in section 4.1, the better skill metrics of GLDAS_CLSM among three CLSM-based RZSM products (GLDAS_CLSM, SMAP L4 and MERRA-2) could be attributed to the more accurate precipitation which is the dominant driver in terrestrial water cycle. GRACE TWS observations were assimilated into GLDAS_CLSM, but previous studies indicated that the assimilation of GRACE TWS has no or little effect on RZSM. This could be attributed to the faster response of soil moisture to atmospheric forcing than groundwater (ZAITCHIK et al., 2008; Houborg et al., 2012; Girotto et al., 2015), the short in situ data record or insufficient spatial sampling (Li et al., 2012). Tian et al. (2016) and Tangdamrongsub et al. (2019) jointly assimilated terrestrial water storage (GRACE TWS) and SSM products. The soil moisture-only assimilation improved the performance of soil moisture estimates against in situ measurements but degraded that of groundwater estimates. The GRACE-only assimilation only enhanced the skill metrics of groundwater."

Reference:

Zaitchik, B. F., Rodell, M. and Reichle, R. H.: Assimilation of GRACE Terrestrial Water Storage Data into a Land Surface Model: Results for the Mississippi River Basin, Journal of Hydrometeorology, 9, 535-548, https://doi.org/10.1175/2007jhm951.1, 2008.

Houborg, R., Rodell, M., Li, B., Reichle, R. and Zaitchik, B. F.: Drought indicators based on model-assimilated Gravity Recovery and Climate Experiment (GRACE) terrestrial water storage observations, Water Resources Research, 48, https://doi.org/10.1029/2011wr011291, 2012.

Li, B., Rodell, M., Zaitchik, B. F., Reichle, R. H., Koster, R. D. and van Dam, T. M.: Assimilation of GRACE terrestrial water storage into a land surface model: Evaluation and potential value for drought monitoring in western and central Europe, Journal of Hydrology, 446-447, 103-115, https://doi.org/10.1016/j.jhydrol.2012.04.035, 2012.

Girotto, M., De Lannoy, G. J. M., Reichle, R. H. and Rodell, M.: Assimilation of gridded terrestrial water storage observations from GRACE into a land surface model, Water Resources Research, 52, 4164-4183, https://doi.org/10.1002/2015wr018417, 2016.

Tian, S., Tregoning, P., Renzullo, L. J., van Dijk, A. I. J. M., Walker, J. P., Pauwels, V. R. N. and Allgeyer, S.: Improved water balance component estimates through joint assimilation of GRACE water storage and SMOS soil moisture retrievals, Water Resources Research, 53, 1820-1840, https://doi.org/10.1002/2016wr019641, 2017.

Tangdamrongsub, N., Han, S.-C., Yeo, I.-Y., Dong, J., Steele-Dunne, S. C., Willgoose, G. and Walker, J. P.: Multivariate data assimilation of GRACE, SMOS, SMAP measurements for improved regional soil moisture and groundwater storage estimates, Advances in Water Resources, 135, https://doi.org/10.1016/j.advwatres.2019.103477, 2020.

6) There is no reference for the in situ measurements other than that "data are available upon request". This should not be acceptable to HESS. For the study to be reproducible, the in situ measurements used here must be made readily accessible to interested researchers.

Response: We will add the reference for the in situ measurements.

Liu, E.; Zhu, Y.; Lü, H.; Horton, R.; Gou, Q.; Wang, X.; Ding, Z.; Xu, H.; Pan, Y. Estimation and Assessment of the Root Zone Soil Moisture from Near-Surface Measurements over Huai River Basin. Atmosphere 2023, 14, 124. https://doi.org/10.3390/atmos14010124.

We will declare the data access in Data availability.

"The soil moisture observations in Huai River Basin are not publicly available but could be requested from the Huaihe River Commission of the Ministry of Water Resources, P. R. C. (https://hrc.gov.cn)."

We will provide the measurements for a subset of 10 stations (see below).

| Station ID | Station Name | Longitude (E) | Latitude (N) | Elevation (m) |
|---|---|---|---|---|
| 50402241 | Taolaoba | 117.164 | 32.184 | 48 |
| 50426801 | Xiaji | 116.540 | 32.654 | 25 |
| 50637450 | Sanshilipu | 116.106 | 32.697 | 27 |
| 50637459 | Xiaqiao | 116.384 | 32.643 | 26 |
| 50903541 | Xibakou | 117.867 | 33.150 | 11 |
| 50907801 | Xulouzha | 116.750 | 33.917 | 30 |
| 50908001 | Suxianzha | 117.083 | 33.667 | 28 |
| 50909701 | Gukouzha | 116.450 | 34.267 | 39 |
| 50924801 | Baoji | 117.113 | 33.158 | 22 |
| 50925801 | Dinghouying | 117.338 | 33.457 | 24 |

7) The discussion in section 5.4 is rambling and does not add useful information. All land models underpinning the products are "1-dimensional" models in the sense that grid cells (or computation elements) are not coupled horizontally other than through the forcing data. The authors make a mess of this simple fact by selectively pointing this out for some products (e.g., for CSLM in Line 506 and for HTESSEL in Line 511) but not others (e.g., Noah, Lines 509-510). The entire section is worthless and should be deleted.

Response: We will delete this section in the revised manuscript.

Minor comments:

1) English style and grammar need some attention. Overall, the paper is readable but there are occasional English language errors.

Response: We do our best to improve the English in the revised manuscript.

2) The graphics generally lack clarity and quality.

a. The axes labels (and other text within the graphics) are often too small to be readable (e.g., Figs 3 and 11)

Response: We will improve the figures in the revised manuscript.

b. Color/symbols are sometimes difficult to distinguish (e.g., Figs 2 and 9)

Response: We will improve the figures in the revised manuscript.

c. The resolution of the images embedded in the pdf is generally poor (e.g., Figs 2, 5, 7)

Response: We will improve the figures in the revised manuscript.

3) Inconsistent use of product names ("SMAP" vs "SMAP L4") (e.g., lines 322-323; Fig 7h label)

Response: We will keep them consistent in the revised manuscript.

4) Line 337/Figure 3: Is the "nonoutlier minimum Q1 – 1.5 x (Q3 – Q1)" meant to reflect the "whiskers" of the box plot? If so, I don't see how this can be correct. Same for the "nonoutlier maximum" (Line 338)

Response: The five horizontal lines of the box plot represent the $0^{th}$ percentile, $25^{th}$ percentile,

50th percentile, 75th percentile and 100th percentile, from bottom to top. We will change the caption of Figure 3.

5) Line 21: "The underestimated SMOS L3 SSM associated with low physical temperature triggers…" This is unclear. Do you mean "The underestimation of SMOS L3 SSM associated with…", or, perhaps more clearly, "The underestimation of the SMOS L3 SSM during cold conditions…" ???

Response: The underestimation of SMOS L3 SSM leads to the underestimation of RZSM in SMOS L4. The underestimation of SMOS L3 SSM is caused by the underestimated physical temperature (not during cold conditions). Maybe "The underestimation of SMOS L3 SSM caused by underestimated physical surface temperature triggers the underestimation of RZSM in SMOS L4" is more clear. We will reword the sentence.

6) Table 3 does not need a column for "Soil layer" because it is the same for each product.

Response: We will remove this column in the revised manuscript and add the soil layer info into the caption.

7) Table 3 needs units for the bias. m3/m3 ??

Response: We will add the unit $m^3/m^3$ in the revised manuscript.

8) Table 3: What is the "Bias (anomaly)" in the final column? Isn't this zero by construction?

Response: The Bias (anomaly) represents the difference between monthly anomaly of RZSM products and monthly anomaly of in situ measurements (refer to section 3.3). It isn't zero because the monthly anomaly of RZSM products and monthly anomaly of in situ measurements are not equal.

9) Line 370: "histograms of normalized RZSM": How was RZSM normalized? I could not find this information.

Response: It is a mistake, we will delete it. In a previous version of the manuscript, we compared the normalized RZSM of different products (Fig. 5). In this manuscript, we use the raw RZSM instead of normalized RZSM.

10) Lines 101-103: The average annual precipitation in the Huai basin is listed as 888 mm precip and the average annual evaporation is listed as ranging from 900 to 1500 mm. A relatively small fraction of the Huai basin is irrigated. How can the average evaporation exceed precipitation by that much?

Response: The cultivated area in the Huai River Basin is approximately 127200 $km^2$, 76% (96667 $km^2$) of which is irrigated according to the Manual of the Huai River Basin Irrigation Area (Chapter 2.1) and Summary of Flood Control Planning for the Huai River Basin (https://hrc.gov.cn). The major water-source infrastructure includes reservoirs, electromechanical wells, diversion sluices and pump stations built along lakes and rivers. Most croplands are irrigated by irrigation channels or a combination of wells and channels (Wang et al., 2021). The heavy irrigation in Huai River Basin could explain the extra water available for evaporation.

Reference

Wang, X., Lü, H., Crow, W. T., Zhu, Y., Wang, Q., Su, J., Zheng, J. and Gou, Q.: Assessment of SMOS and SMAP soil moisture products against new estimates combining physical model, a statistical model, and in-situ observations: A case study over the Huai River Basin, China, J. Hydro., 598, 126468, https://doi.org/10.1016/j.jhydrol.2021.126468, 2021.